# FUNCTIONAL VS. PARAMETRIC EQUIVALENCE OF ReLU NETWORKS

**Mary Phuong & Christoph H. Lampert**
IST Austria
Am Campus 1, Klosterneuburg, Austria
{bphuong,chl}@ist.ac.at

We address the following question: How redundant is the parameterisation of ReLU networks? Specifically, we consider transformations of the weight space which leave the function implemented by the network intact. Two such transformations are known for feed-forward architectures: permutation of neurons within a layer, and positive scaling of all incoming weights of a neuron coupled with inverse scaling of its outgoing weights. In this work, we show for architectures with non-increasing widths that permutation and scaling are in fact the *only* function-preserving weight transformations. For any eligible architecture we give an explicit construction of a neural network such that any other network that implements the same function can be obtained from the original one by the application of permutations and rescaling. The proof relies on a geometric understanding of boundaries between linear regions of ReLU networks, and we hope the developed mathematical tools are of independent interest.

## 1 INTRODUCTION

Ever since its early successes, deep learning has been a puzzle for machine learning theorists. Multiple aspects of deep learning seem at first sight to contradict common sense: single-hidden-layer networks suffice to approximate any continuous function (Cybenko, 1989; Hornik et al., 1989), yet in practice deeper is better; the loss surface is highly non-convex, yet it can be minimised by first-order methods; the capacity of the model class is immense, yet deep networks tend not to overfit (Zhang et al., 2017).

Recent investigations into these and other questions have emphasised the role of over-parameterisation, or highly redundant function representation. It is now known that over-parameterised networks enjoy both easier training (Allen-Zhu et al., 2019; Du et al., 2019; Frankle & Carbin, 2019), and better generalisation (Belkin et al., 2019; Neyshabur et al., 2019; Novak et al., 2018). However, the specific mechanism by which over-parameterisation operates is still largely a mystery.

In this work, we study one particular aspect of over-parameterisation, namely the ability of neural networks to represent a target function in many different ways. In other words, we ask whether many different parameter configurations can give rise to the same function. Such a notion of parameterisation redundancy has so far remained unexplored, despite its potential connections to the structure of the loss landscape, as well as to the literature on neural network capacity in general.

Specifically, we consider feed-forward ReLU networks, with weight matrices $\mathbf{W}_1, \ldots, \mathbf{W}_L$, and biases $\mathbf{b}_1, \ldots, \mathbf{b}_L$. We study parameter transformations which preserve the output behaviour of the network $h(\mathbf{z}) = \mathbf{W}_L \sigma(\mathbf{W}_{L-1}\sigma(\ldots \mathbf{W}_1\mathbf{z} + \mathbf{b}_1 \ldots) + \mathbf{b}_{L-1}) + \mathbf{b}_L$ for all inputs $\mathbf{z}$ in some domain $\mathcal{Z}$. Two such transformations are known for feed-forward ReLU architectures:

1. Permutation of units (neurons) within a layer, i.e. for some permutation matrix $\mathbf{P}$,

$$\mathbf{W}_l \leftarrow \mathbf{P}\mathbf{W}_l, \qquad\qquad \mathbf{b}_l \leftarrow \mathbf{P}\mathbf{b}_l, \qquad\qquad (1)$$

$$\mathbf{W}_{l+1} \leftarrow \mathbf{W}_{l+1}\mathbf{P}^{-1}. \qquad\qquad (2)$$

2. Positive scaling of all incoming weights of a unit coupled with inverse scaling of its outgoing weights. Applied to a whole layer, with potentially different scaling factors arranged into a diagonal matrix $\mathbf{M}$, this can be written as

$$\mathbf{W}_l \leftarrow \mathbf{M}\mathbf{W}_l, \qquad\qquad \mathbf{b}_l \leftarrow \mathbf{M}\mathbf{b}_l, \qquad\qquad (3)$$

$$\mathbf{W}_{l+1} \leftarrow \mathbf{W}_{l+1}\mathbf{M}^{-1}. \qquad\qquad (4)$$

Our main theorem applies to architectures with non-increasing widths, and shows that *there are no other function-preserving parameter transformations besides permutation and scaling.* Stated formally:

**Theorem 1.** *Consider a bounded open nonempty domain $\mathcal{Z} \subseteq \mathbb{R}^{d_0}$ and any architecture $(d_0, \ldots, d_L)$ with $d_0 \geq d_1 \geq \cdots \geq d_{L-1} \geq 2$, $d_L = 1$. For this architecture, there exists a ReLU network $h_{\boldsymbol{\theta}} : \mathcal{Z} \to \mathbb{R}$, or equivalently a setting of the weights $\boldsymbol{\theta} \triangleq (\mathbf{W}_1, \mathbf{b}_1, \ldots, \mathbf{W}_L, \mathbf{b}_L)$, such that for any 'general' ReLU network $h_{\boldsymbol{\eta}} : \mathcal{Z} \to \mathbb{R}$ (with the same architecture) satisfying $h_{\boldsymbol{\theta}}(\mathbf{z}) = h_{\boldsymbol{\eta}}(\mathbf{z})$ for all $\mathbf{z} \in \mathcal{Z}$, there exist permutation matrices $\mathbf{P}_1, \ldots \mathbf{P}_{L-1}$, and positive diagonal matrices $\mathbf{M}_1, \ldots, \mathbf{M}_{L-1}$, such that*

$$
\begin{aligned}
\mathbf{W}_1 &= \mathbf{M}_1 \mathbf{P}_1 \mathbf{W}'_1, & \mathbf{b}_1 &= \mathbf{M}_1 \mathbf{P}_1 \mathbf{b}'_1, \\
\mathbf{W}_l &= \mathbf{M}_l \mathbf{P}_l \mathbf{W}'_l \mathbf{P}_{l-1}^{-1} \mathbf{M}_{l-1}^{-1}, & \mathbf{b}_l &= \mathbf{M}_l \mathbf{P}_l \mathbf{b}'_l, & l \in \{2, \ldots, L-1\}, \\
\mathbf{W}_L &= \mathbf{W}'_L \mathbf{P}_{L-1}^{-1} \mathbf{M}_{L-1}^{-1}, & \mathbf{b}_L &= \mathbf{b}'_L,
\end{aligned} \tag{5}
$$

*where $\boldsymbol{\eta} \triangleq (\mathbf{W}'_1, \mathbf{b}'_1, \ldots, \mathbf{W}'_L, \mathbf{b}'_L)$ are the parameters of $h_{\boldsymbol{\eta}}$.*

In the above, 'general' networks is a class of networks meant to exclude degenerate cases. We give a more precise definition in Section 3; for now it suffices to note that almost all networks are general.

The proof of the result relies on a geometric understanding of prediction surfaces of ReLU networks. These surfaces are piece-wise linear functions, with non-differentiabilities or 'folds' between linear regions. It turns out that folds carry a lot of information about the parameters of a network, so much in fact, that some networks are uniquely identified (up to permutation and scaling) by the function they implement. This is the main insight of the theorem.

In the following sections, we introduce in more detail the concept of a fold-set, and describe its geometric structure for a subclass of ReLU networks. The paper culminates in a proof sketch of the main result. The full proof, including proofs of intermediate results, is included in the Appendix.

## 2 RELATED WORK

The functional equivalence of neural networks is a well-researched topic in classical connectionist literature. The problem was first posed by Hecht-Nielsen (1990), and soon resolved for feed-forward networks with the tanh activation function by Chen et al. (1993), who showed that any smooth transformation of the weight space that preserves the function of all neural networks is necessarily a composition of permutations and sign flips. For the same class of networks, Fefferman & Markel (1994) showed a somewhat stronger result: knowledge of the input-output mapping of a neural network determines both its architecture and its weights, up to permutations and sign flips. Similar results have been proven for single-layer networks with a saturating activation function such as sigmoid or RBF (Kůrková & Kainen, 1994), as well as single-layer recurrent networks with a smooth activation function (Albertini & Sontag, 1993a;b).

To the best of our knowledge, no such theoretical results exist for networks with the ReLU activation, which is non-saturating, asymmetric and non-smooth. Broadly related is the recent work by Petersen et al. (2018) and Berner et al. (2019) who study whether two neural networks (ReLU or otherwise) that are close in the functional space have parameterisations that are close in the weight space. This is called inverse stability. In contrast, we are interested in ReLU networks that are functionally identical, and ask about all their possible parameterisations.

In terms of proof technique, our approach is based on the geometry of piece-wise linear functions, specifically the boundaries between linear regions. The intuition for this kind of analysis has previously been presented by Raghu et al. (2017) and Serra et al. (2018), and somewhat similar proof techniques to ours have been used by Hanin & Rolnick (2019) in the context of counting the number of linear decision regions.

Finally, the sets of equivalent parametrisations can be viewed as symmetries in the weight space, with implications for optimisation. Multiple authors, including e.g. Neyshabur et al. (2015); Badrinarayanan et al. (2016); Stock et al. (2019), have observed that the naive loss gradient is sensitive to reparametrisation by scaling, and proposed alternative, scaling-invariant optimisation procedures.

## 3 ReLU networks

This section introduces notation and two important classes of ReLU networks that we refer to throughout the manuscript. We denote by $\sigma$ the ReLU function: $\sigma(\mathbf{u})_i = \max\{0, u_i\}$ for $i \in [\dim(\mathbf{u})]$; the subscript index denotes the corresponding vector element.

**ReLU network.** Let $\mathcal{Z} \subseteq \mathbb{R}^{d_0}$ with $d_0 \geq 2$ be a nonempty open set, and let $\boldsymbol{\theta} \triangleq (\mathbf{W}_1, \mathbf{b}_1, \dots, \mathbf{W}_L, \mathbf{b}_L)$ be the network's parameters, with $\mathbf{W}_l \in \mathbb{R}^{d_l \times d_{l-1}}$, $\mathbf{b}_l \in \mathbb{R}^{d_l}$, and $d_L = 1$. We denote the corresponding ReLU network by $h_{\boldsymbol{\theta}} : \mathcal{Z} \to \mathbb{R}$, where

$$h_{\boldsymbol{\theta}} \triangleq h_{\boldsymbol{\theta}}^L \circ \sigma \circ h_{\boldsymbol{\theta}}^{L-1} \circ \cdots \circ \sigma \circ h_{\boldsymbol{\theta}}^1, \tag{6}$$

and $h_{\boldsymbol{\theta}}^l(\mathbf{z}) = \mathbf{W}_l \cdot \mathbf{z} + \mathbf{b}_l$. For $1 \leq l \leq k \leq L$, we also introduce notation for truncated networks,

$$h_{\boldsymbol{\theta}}^{l:k} \triangleq h_{\boldsymbol{\theta}}^k \circ \sigma \circ h_{\boldsymbol{\theta}}^{k-1} \cdots \circ \sigma \circ h_{\boldsymbol{\theta}}^l. \tag{7}$$

We will omit the subscript $\boldsymbol{\theta}$ when it is clear from the context.

**General ReLU network.** In this work, we restrict our attention to so-called *general* ReLU networks. Intuitively, a general network is one that satisfies a number of non-degeneracy properties, such as all weight matrices having non-zero entries and full rank, no two network units exactly cancelling each other out, etc. It can be shown[1] that *almost all* ReLU networks are general. In other words, a sufficient condition for a ReLU network to be general with probability one is that its weights are sampled from a distribution with a density.

More formally, a general ReLU network is one that satisfies the following three conditions.

1. For any unit $(l, i)$, the local optima of $h_i^{1:l}$ do *not* have value exactly zero.
2. For all $k \leq l$ and all diagonal matrices $(\mathbf{I}_k, \dots, \mathbf{I}_l)$ with entries in $\{0, 1\}$,

   $$\mathrm{rank}(\mathbf{I}_l \mathbf{W}_l \mathbf{I}_{l-1} \cdots \mathbf{I}_k \mathbf{W}_k) = \min\{d_{k-1}, \mathrm{rank}(\mathbf{I}_k), \dots, \mathrm{rank}(\mathbf{I}_{l-1}), \mathrm{rank}(\mathbf{I}_l)\}. \tag{8}$$

3. For any two units $(l, i)$, $(k, j)$, any linear region $\mathcal{R}_1 \subseteq \mathcal{Z}$ of $h_i^{1:l}$, and any linear region $\mathcal{R}_2 \subseteq \mathcal{Z}$ of $h_j^{1:k}$, the linear functions implemented by $h_i^{1:l}$ on $\mathcal{R}_1$ and $h_j^{1:k}$ on $\mathcal{R}_2$ are *not* multiples of each other.

General networks are convenient to study, as they exclude many degenerate special cases.

The second important class of ReLU networks are so-called *transparent* networks. Their significance as well as their name will become clear in the next section. For now, we state the definition.

**Transparent ReLU network.** A ReLU network $h : \mathcal{Z} \to \mathbb{R}$ is called transparent if for all $\mathbf{z} \in \mathcal{Z}$ and $l \in [L-1]$, there exists $i \in [d_l]$ such that $h_i^{1:l}(\mathbf{z}) \geq 0$. In words, we require that for any input, at least one unit on each layer is active.

## 4 Fold-sets

In this section we introduce the concept of *fold-sets*, which is key to our understanding of ReLU networks and their prediction surfaces. Since ReLU networks are piece-wise linear functions, a great deal about them is revealed by the boundaries between individual linear regions. A network's fold-set is simply the union of all these boundaries.

More formally, if $\mathcal{Z}$ is an open set, and $f : \mathcal{Z} \to \mathbb{R}$ is any continuous, piece-wise linear function, we define the *fold-set of $f$*, denoted by $\mathcal{F}(f)$, as the set of all points at which $f$ is non-differentiable.

It turns out there is a class of networks whose fold-sets are especially easy to understand; these are the ones we have termed *transparent*. For transparent networks, we have the following characterisation of the fold-set (which also motivates the name 'transparent').

---

[1]See Appendix, Lemmas A.10, A.11 and A.12.

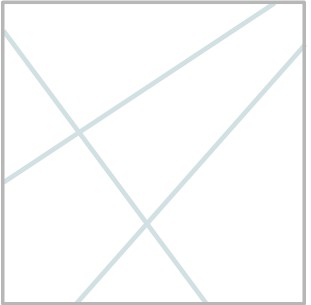 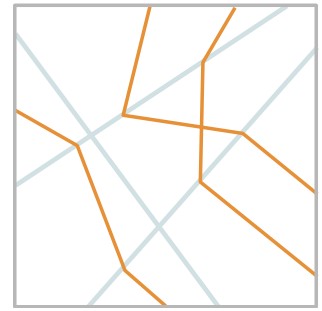 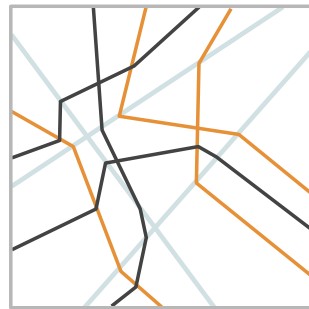

Figure 1: A piece-wise linear surface of order one, two and three.[3]

**Lemma 1.** *If $h : \mathcal{Z} \to \mathbb{R}$ is a general and transparent ReLU network, then*

$$\mathcal{F}(h) = \bigcup_{l,i} \big\{ \mathbf{z} \mid h_i^{1:l}(\mathbf{z}) = 0 \big\}. \tag{9}$$

To appreciate the significance of the lemma, suppose we are given some transparent ReLU network function $h$ and we want to infer its parameters. This lemma shows that the knowledge of the *end-to-end mapping* $h \triangleq h^{1:L}$ in fact gives us information about the network's *hidden units* $h_i^{1:l}$ (hence 'transparent'). Moreover, this information is very explicit: we observe the units' zero-level sets, which in the case of a linear unit on a full-dimensional space already determines the unit's parameters up to scaling[2]. Of course, dealing with piece-wise linearity and disambiguating the union into its constituent zero-level sets remains a challenge for upcoming sections.

## 5 PIECE-WISE LINEAR SURFACES

In this section, we provide a geometric description of fold-sets of transparent networks. Intuitively, the fold-sets look like the sets shown in Figure 1. The first-layer units of a network are linear, so the component $\bigcup_i \big\{ \mathbf{z} \mid h_i^{1:1}(\mathbf{z}) = 0 \big\}$ of the fold-set (9) is a union of hyperplanes, illustrated by the blue lines in Figure 1. These hyperplanes partition the input space into a number of regions that each correspond to a different activation pattern. For a fixed activation pattern, or equivalently on each region, the second-layer units are linear, so their zero-level sets $\bigcup_i \big\{ \mathbf{z} \mid h_i^{1:2}(\mathbf{z}) = 0 \big\}$ are composed of piece-wise hyperplanes on the partition induced by the first-layer units. This is shown by the orange lines in Figure 1. More generally, the $l^{\text{th}}$-layer zero-level sets $\bigcup_i \big\{ \mathbf{z} \mid h_i^{1:l}(\mathbf{z}) = 0 \big\}$ consist of piece-wise hyperplanes on the partition induced by all lower-layer units. This yields a fold-set that looks like the set in the right pane of Figure 1, but potentially much more complicated.

We now define these concepts more precisely.

**Piece-wise hyperplane.** Let $\mathcal{P}$ be a partition of $\mathcal{Z}$. We say $\mathcal{H} \subseteq \mathcal{Z}$ is a *piece-wise hyperplane* with respect to partition $\mathcal{P}$, if $\mathcal{H}$ is nonempty and there exist $(\mathbf{w}, b) \neq (\mathbf{0}, 0)$ and $P \in \mathcal{P}$ such that $\mathcal{H} = \{ \mathbf{z} \in P \mid \mathbf{w}^\mathsf{T} \mathbf{z} + b = 0 \}$.

**Piece-wise linear surface.** A set $\mathcal{S} \subseteq \mathcal{Z}$ is called a *piece-wise linear surface on $\mathcal{Z}$ of order $\kappa$* if it has a representation of the form $\mathcal{S} = \bigcup_{l \in [\kappa], i \in [n_l]} \mathcal{H}_i^l$, where each $\mathcal{H}_i^l$ is a piece-wise hyperplane with respect to the partition induced by $\bigcup_{k \in [l-1], j \in [n_k]} \mathcal{H}_j^k$, and no number smaller than $\kappa$ admits such a representation.

Using these definitions, the following lemma formalises the intuition behind Figure 1.

**Lemma 2.** *If $h$ is a general and transparent ReLU network, then its fold-set is a piece-wise linear surface of order at most $L - 1$.*

---

[2]See Appendix, Lemma A.19.

[3]A similar figure has appeared in the work of Raghu et al. (2017).

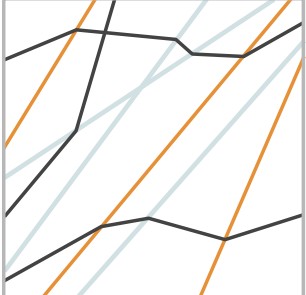 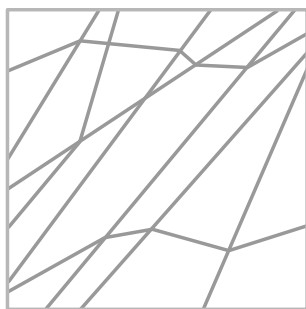 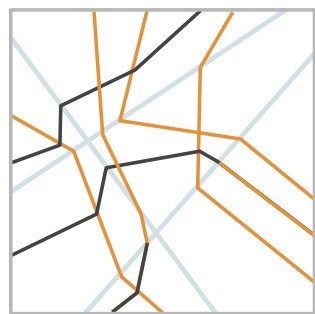

Figure 2: A piece-wise linear surface with few intersections between piece-wise hyperplanes. From the fold-set alone (right) it is not possible to determine if a hyperplane emerged from the first layer (left, blue) or from the second one (left, orange).

Figure 3: Greedy layer assignment to the piece-wise linear surface in Figure 1.

The final ingredient we will need to be able to reason about the parameterisation of ReLU networks is a more precise characterisation of the fold-set, in particular, the *dependence structure* between individual piece-wise hyperplanes. For example, consider the piece-wise linear surface in Figure 1 and compare it to the one in Figure 2. Suppose as before that the blue hyperplanes come from first-layer units, the orange hyperplanes come from second-layer units, and the black hyperplanes come from third-layer units. The difference between Figure 1 and Figure 2 is that if we observe only the fold-set, i.e. only the union of the zero-level sets over all layers (as shown in the right pane of Figure 2), then in the case of Figure 2, it is impossible to know which folds come from which layers. For instance, the blue folds and the orange folds could be assigned to the first and second layer almost arbitrarily; there is not enough information (i.e. intersection) in the fold-set to tell which is which. In contrast, the piece-wise linear surface in the right pane of Figure 1 could in principle be disambiguated into first-, second- and third- layer folds by the following procedure:

1. Take the largest possible union of hyperplanes that is a subset of the fold-set, and assign the hyperplanes to layer one.

2. Take all piece-wise hyperplanes with respect to the partition induced by the first-layer folds, and assign them to layer two.

3. Take all piece-wise hyperplanes with respect to the partition induced by the first- and second- layer folds, and assign them to layer three.

This procedure is not guaranteed to assign all folds to their original layers because it ignores how piece-wise hyperplanes are connected; for example for the piece-wise linear surface in Figure 1, the procedure yields the layer assignment shown in Figure 3. However, it is sufficient for our purposes, and it is easier to work with mathematically.

Formally, for a piece-wise linear surface $\mathcal{S}$, we denote

$$\square_k \mathcal{S} := \bigcup \{\mathcal{S}' \subseteq \mathcal{S} \mid \mathcal{S}' \text{ is a piece-wise linear surface of order at most } k\}. \tag{10}$$

One can show[4] that $\square_k \mathcal{S}$ is itself a piece-wise linear surface of order at most $k$, so one can think of $\square_k \mathcal{S}$ as the 'largest possible' subset of $\mathcal{S}$ that is a piece-wise linear surface of order at most $k$. For the piece-wise linear surface in Figure 3, the set $\square_1 \mathcal{S}$ consists of the blue hyperplanes, $\square_2 \mathcal{S}$ consists of the blue and the orange (piece-wise) hyperplanes, and $\square_3 \mathcal{S} = \mathcal{S}$.

This definition allows us to uniquely decompose $\mathcal{S}$ into its piece-wise hyperplanes. Let $\mathcal{S} = \bigcup_{l \in [\kappa], i \in [n_l]} \mathcal{H}_i^l$ be any representation of $\mathcal{S}$ in terms of its piece-wise hyperplanes. We say the representation is *canonical* if each $\mathcal{H}_i^l$ is distinct and $\bigcup_{l \in [k], i \in [n_l]} \mathcal{H}_i^l = \square_k \mathcal{S}$ for all $k \in [\kappa]$. One can show[5] that such a representation exists and is unique up to subscript indexing. Importantly, it assigns a unique 'layer' to each piece-wise hyperplane, its superscript.

---

[4]See Appendix, Lemma A.1.
[5]See Appendix, Lemmas A.5 and A.6.

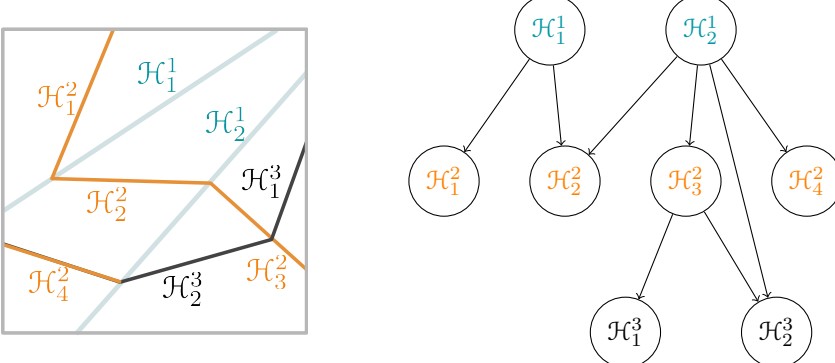

Figure 4: A piece-wise linear surface in canonical form and its dependency graph.

The *dependency graph* (see also Figure 4) is a way to formally describe the dependencies between piece-wise hyperplanes.

**Dependency graph.** Let $\mathcal{S} = \bigcup_{l \in [\kappa], i \in [n_l]} \mathcal{H}_i^l$ be the canonical representation of $\mathcal{S}$. The *dependency graph of $\mathcal{S}$* is the directed graph that has the piece-wise hyperplanes $\{\mathcal{H}_i^l\}_{l,i}$ as vertices, and has an edge $\mathcal{H}_i^l \to \mathcal{H}_j^k$ iff $l < k$ and $\operatorname{relint} \mathcal{H}_i^l \cap \operatorname{cl} \mathcal{H}_j^k \neq \emptyset$. That is, there is and edge $\mathcal{H}_i^l \to \mathcal{H}_j^k$ if $\mathcal{H}_j^k$ 'depends on' or 'bends at' $\mathcal{H}_i^l$.

## 6 MAIN RESULT

With all the necessary concepts in place, we now put the pieces together and explain the proof idea behind the main result. We restate the theorem here for the reader's convenience.

**Theorem 1.** *Consider a bounded open nonempty domain $\mathcal{Z} \subseteq \mathbb{R}^{d_0}$ and any architecture $(d_0, \ldots, d_L)$ with $d_0 \geq d_1 \geq \cdots \geq d_{L-1} \geq 2$, $d_L = 1$. For this architecture, there exists a ReLU network $h_{\boldsymbol{\theta}} : \mathcal{Z} \to \mathbb{R}$ such that for any general ReLU network $h_{\boldsymbol{\eta}} : \mathcal{Z} \to \mathbb{R}$ (with the same architecture) satisfying $h_{\boldsymbol{\theta}}(\mathbf{z}) = h_{\boldsymbol{\eta}}(\mathbf{z})$ for all $\mathbf{z} \in \mathcal{Z}$, there exist permutation matrices $\mathbf{P}_1, \ldots \mathbf{P}_{L-1}$, and positive diagonal matrices $\mathbf{M}_1, \ldots, \mathbf{M}_{L-1}$, such that*

$$
\begin{aligned}
\mathbf{W}_1 &= \mathbf{M}_1 \mathbf{P}_1 \mathbf{W}_1', & \mathbf{b}_1 &= \mathbf{M}_1 \mathbf{P}_1 \mathbf{b}_1', & \\
\mathbf{W}_l &= \mathbf{M}_l \mathbf{P}_l \mathbf{W}_l' \mathbf{P}_{l-1}^{-1} \mathbf{M}_{l-1}^{-1}, & \mathbf{b}_l &= \mathbf{M}_l \mathbf{P}_l \mathbf{b}_l', & l \in \{2, \ldots, L-1\}, \\
\mathbf{W}_L &= \mathbf{W}_L' \mathbf{P}_{L-1}^{-1} \mathbf{M}_{L-1}^{-1}, & \mathbf{b}_L &= \mathbf{b}_L', &
\end{aligned}
\tag{11}
$$

*where $(\mathbf{W}_1, \mathbf{b}_1, \ldots, \mathbf{W}_L, \mathbf{b}_L)$ are the parameters of $h_{\boldsymbol{\theta}}$, and $(\mathbf{W}_1', \mathbf{b}_1', \ldots, \mathbf{W}_L', \mathbf{b}_L')$ are the parameters of $h_{\boldsymbol{\eta}}$.*

In other words, for architectures with non-increasing widths, there exists a ReLU network $h$ such that knowledge of the input-output mapping $h$ determines the network's parameters uniquely up to permutation and scaling.

The idea behind the proof is as follows. Suppose we are given the function $h$. Then we also know its fold-set $\mathcal{F}(h)$, and if $h$ is general and transparent, the fold-set is a piece-wise linear surface (by Lemma 2) of the form $\mathcal{F}(h) = \bigcup_{l,i} \{\mathbf{z} \mid h_i^{1:l}(\mathbf{z}) = 0\}$. As we have mentioned earlier, this union of zero-level sets contains a lot of information about the network's parameters, provided we can disambiguate the union to obtain the zero-level sets of individual units.

This disambiguation of the union is crucial, but is impossible in general. To see why, consider the first-layer units: given $\mathcal{F}(h)$, we want to identify $\bigcup_i \{\mathbf{z} \mid h_i^{1:1}(\mathbf{z}) = 0\}$. We know that $\bigcup_i \{\mathbf{z} \mid h_i^{1:1}(\mathbf{z}) = 0\}$ is a union of $d_1$ hyperplanes that is a subset of $\square_1 \mathcal{F}(h)$, so if $\square_1 \mathcal{F}(h)$ is a union of $d_1$ hyperplanes, we are done. In general however, $\mathcal{F}(h)$ may contain more than $d_1$ hyperplanes, such as for example in Figure 2. In such a setting it is impossible to tell which hyperplanes come from the first layer.

The key insight here is the following: even though, say, a last-layer unit can create a fold that looks like a hyperplane, this hyperplane cannot have any dependencies, or descendants in the dependency graph. This follows from the fact that the layer is the last. More generally, if a (piece-wise) hyperplane has a chain of descendants of length $m$, it must come from a layer that is at least $m$ layers below the last one. Formally, we have the following lemma.

**Lemma 3.** *Let $h : \mathcal{Z} \to \mathbb{R}$ be a general ReLU network. Denote $\mathcal{S} := \bigcup_{l \in [\lambda], i \in [d_l]} \left\{ \mathbf{z} \mid h_i^{1:l}(\mathbf{z}) = 0 \right\}$ and let $\mathcal{S} = \bigcup_{k \in [\kappa], j \in [n_k]} \mathcal{H}_j^k$ be the canonical representation of $\mathcal{S}$. Then for all $\mathcal{H}_j^k$ there exists a unit $(l, i)$ with $l \geq k$ such that $\mathcal{H}_j^k \subseteq \left\{ \mathbf{z} \mid h_i^{1:l}(\mathbf{z}) = 0 \right\}$. Moreover, if the dependency graph of $\mathcal{S}$ contains a directed path of length $m$ starting at $\mathcal{H}_j^k$, then $l \leq \lambda - m$.*

**Main proof idea.** This lemma motivates the main idea of the proof. We explicitly construct a network $h$ such that the dependency graph of its fold-set is well connected. More precisely, we ensure that each of the hyperplanes corresponding to first-layer units has a chain of descendants of length $L - 2$. This implies by Lemma 3 that the first-layer hyperplanes can be identified as such, using only the information contained in the fold-set. One can show that this is sufficient to recover the parameters $\mathbf{W}_1, \mathbf{b}_1$, up to permutation and scaling. To extend the argument to higher-layers, we then consider the truncated network $h^{l:L}$. In $h^{l:L}$, layer $l$ becomes the first layer, and we apply the same reasoning as above to recover $\mathbf{W}_l, \mathbf{b}_l$.

The next lemma shows that a network with a 'well connected' dependency graph exists. In what follows, $f|_{\mathcal{A}}$ denotes the restriction of a function $f$ to a domain $\mathcal{A}$, and $\mathcal{Z}_{\boldsymbol{\theta}}^l \triangleq \left\{ \sigma(h_{\boldsymbol{\theta}}^{1:l}(\mathbf{z})) \mid \mathbf{z} \in \mathcal{Z} \right\}$ is the set of all possible inputs to the truncated network $h^{l:L}$. For notational convenience, we define $\mathcal{Z}_{\boldsymbol{\theta}}^0 \triangleq \mathcal{Z}$.

**Lemma 4.** *For a bounded open nonempty domain $\mathcal{Z}$ and architecture $(d_0, \ldots, d_L)$ with $d_0 \geq d_1 \geq \cdots \geq d_{L-1} \geq 2$, $d_L = 1$, there exists a general transparent ReLU network $h : \mathcal{Z} \to \mathbb{R}$ such that for $l \in [L - 1]$, the fold-set $\mathcal{F}(h^{l:L}|_{\text{int} \mathcal{Z}^{l-1}})$ is a piece-wise linear surface whose dependency graph contains $d_l$ directed paths of length $(L - 1 - l)$ with distinct starting vertices.*

Theorem 1 then follows by the inductive argument outlined above.

**Proof sketch of Theorem 1.** Let $h_{\boldsymbol{\theta}}$ be the network from Lemma 4. One can show that if $h_{\boldsymbol{\theta}}$ is transparent, and $h_{\boldsymbol{\eta}}(\mathbf{z}) = h_{\boldsymbol{\theta}}(\mathbf{z})$ for all $\mathbf{z} \in \mathcal{Z}$, then also $h_{\boldsymbol{\eta}}$ is transparent, and all the truncated networks $h_{\boldsymbol{\theta}}^{l:L}$, $h_{\boldsymbol{\eta}}^{l:L}$ are transparent.

We proceed by induction. Let $l = 1$. Then we have

$$h_{\boldsymbol{\theta}}^{l:L}\big|_{\text{int} \, \mathcal{Z}_{\boldsymbol{\theta}}^{l-1}} \equiv h_{\boldsymbol{\theta}} \equiv h_{\boldsymbol{\eta}} \equiv h_{\boldsymbol{\eta}}^{l:L}\big|_{\text{int} \, \mathcal{Z}_{\boldsymbol{\theta}}^{l-1}} \tag{12}$$

which implies $\mathcal{F}(h_{\boldsymbol{\theta}}^{l:L}|_{\text{int} \, \mathcal{Z}_{\boldsymbol{\theta}}^{l-1}}) = \mathcal{F}(h_{\boldsymbol{\eta}}^{l:L}|_{\text{int} \, \mathcal{Z}_{\boldsymbol{\theta}}^{l-1}})$. (For notational convenience, we will omit the domain restriction for now.) Because both networks are general and transparent, the fold-sets are representable as unions of the respective zero-level sets, and we obtain

$$\bigcup_{k \in [L-l], j \in [d_k]} \left\{ \mathbf{z} \mid h_{\boldsymbol{\theta}}^{l:l-1+k}[j](\mathbf{z}) = 0 \right\} = \bigcup_{k \in [L-l], j \in [d_k]} \left\{ \mathbf{z} \mid h_{\boldsymbol{\eta}}^{l:l-1+k}[j](\mathbf{z}) = 0 \right\} \tag{13}$$

This is a piece-wise linear surface, whose dependency graph by Lemma 4 contains $d_l$ directed paths of length $(L-1-l)$ with distinct starting vertices. Denote these vertices $\mathcal{H}_1, \ldots, \mathcal{H}_{d_l}$. By Lemma 3, $\mathcal{H}_i \subseteq \left\{ \mathbf{z} \mid h_{\boldsymbol{\theta}}^{l:l-1+\lambda}[\iota](\mathbf{z}) = 0 \right\}$ for some $(\lambda, \iota)$ with $\lambda \leq (L - l) - (L - 1 - l) = 1$. We thus obtain $\bigcup_{i \in [d_l]} \mathcal{H}_i \subseteq \bigcup_{i \in [d_l]} \left\{ \mathbf{z} \mid h_{\boldsymbol{\theta}}^l[\iota](\mathbf{z}) = 0 \right\}$, where on the left-hand side we have a union of $d_l$ hyperplanes, and on the right-hand side we have a union of at most $d_l$ hyperplanes. It follows that the two sides are equal, and by applying the same argument to $h_{\boldsymbol{\eta}}$, we get

$$\bigcup_{i \in [d_l]} \left\{ \mathbf{z} \mid h_{\boldsymbol{\theta}}^l[i](\mathbf{z}) = 0 \right\} = \bigcup_{i \in [d_l]} \left\{ \mathbf{z} \mid h_{\boldsymbol{\eta}}^l[i](\mathbf{z}) = 0 \right\}. \tag{14}$$

Therefore there must exist a permutation $\pi : [d_l] \to [d_l]$ such that

$$\left\{ \mathbf{z} \mid h_{\boldsymbol{\theta}}^l[i](\mathbf{z}) = 0 \right\} = \left\{ \mathbf{z} \mid h_{\boldsymbol{\eta}}^l[\pi(i)](\mathbf{z}) = 0 \right\} \tag{15}$$

for all $i$. One can show[6] that this implies the existence of scalars $m_1, \ldots m_{d_l}$, such that

$$(\mathbf{W}_l[i,:], b_l[i]) = m_i (\mathbf{W}'_l[\pi(i),:], b'_l[\pi(i)]). \tag{16}$$

We know that $m_i \neq 0$ because the folds $\{\mathbf{z} \mid h^l_{\boldsymbol{\theta}}[i](\mathbf{z}) = 0\}$, $\{\mathbf{z} \mid h^l_{\boldsymbol{\eta}}[i](\mathbf{z}) = 0\}$, are nonempty; otherwise $\bigcup_{i \in [d_l]} \mathcal{H}_i$ could not be a union of $d_l$ hyperplanes. We have thus shown that there exists a permutation matrix $\mathbf{P}_l \in \mathbb{R}^{d_l \times d_l}$ and a nonzero-entry diagonal matrix $\mathbf{M}_l \in \mathbb{R}^{d_l \times d_l}$ such that $\mathbf{W}_l = \mathbf{M}_l \mathbf{P}_l \mathbf{W}'_l$ and $\mathbf{b}_l = \mathbf{M}_l \mathbf{P}_l \mathbf{b}'_l$. One can also show that the scalars $m_i$ are positive.[7]

For the inductive step, let $l \in \{2, \ldots, L-1\}$, and assume that there exist permutation matrices $\mathbf{P}_1, \ldots, \mathbf{P}_{l-1}$, and positive-entry diagonal matrices $\mathbf{M}_1, \ldots, \mathbf{M}_{l-1}$, such that (65) holds up to layer $l-1$. Then $h^{1:l-1}_{\boldsymbol{\theta}} \equiv \mathbf{M}_{l-1} \mathbf{P}_{l-1} h^{1:l-1}_{\boldsymbol{\eta}}$. Since the end-to-end mappings are the same, $h^{1:L}_{\boldsymbol{\theta}} \equiv h^{1:L}_{\boldsymbol{\eta}}$, it follows that the truncated mappings satisfy

$$h^{l:L}_{\boldsymbol{\theta}}\big|_{\text{int } \mathcal{Z}^{l-1}_{\boldsymbol{\theta}}} \equiv \left( h^{l:L}_{\boldsymbol{\eta}} \circ \mathbf{P}^{-1}_{l-1} \mathbf{M}^{-1}_{l-1} \right)\big|_{\text{int } \mathcal{Z}^{l-1}_{\boldsymbol{\theta}}} \equiv h^{l:L}_{\tilde{\boldsymbol{\eta}}}\big|_{\text{int } \mathcal{Z}^{l-1}_{\boldsymbol{\theta}}}, \tag{17}$$

where $\tilde{\boldsymbol{\eta}} := (\mathbf{W}'_l \mathbf{P}^{-1}_{l-1} \mathbf{M}^{-1}_{l-1}, \mathbf{b}'_l, \mathbf{W}'_{l+1}, \mathbf{b}'_{l+1}, \ldots, \mathbf{W}'_L, \mathbf{b}'_L)$. We therefore apply the same argument to $h^{l:L}_{\boldsymbol{\theta}}\big|_{\text{int } \mathcal{Z}^{l-1}_{\boldsymbol{\theta}}}$ and $h^{l:L}_{\tilde{\boldsymbol{\eta}}}\big|_{\text{int } \mathcal{Z}^{l-1}_{\boldsymbol{\theta}}}$ as we presented above for the case $l = 1$. We obtain that there exists a permutation matrix $\mathbf{P}_l \in \mathbb{R}^{d_l \times d_l}$ and a positive-entry diagonal matrix $\mathbf{M}_l \in \mathbb{R}^{d_l \times d_l}$ such that

$$\mathbf{W}_l = \mathbf{M}_l \mathbf{P}_l \mathbf{W}'_l \mathbf{P}^{-1}_{l-1} \mathbf{M}^{-1}_{l-1}, \qquad \mathbf{b}_l = \mathbf{M}_l \mathbf{P}_l \mathbf{b}'_l. \tag{18}$$

Finally, consider the last layer. We know that $h^{1:L-1}_{\boldsymbol{\theta}} \equiv \mathbf{M}_{L-1} \mathbf{P}_{L-1} h^{1:L-1}_{\boldsymbol{\eta}}$, which implies $h^L_{\boldsymbol{\theta}} \equiv h^L_{\boldsymbol{\eta}} \circ \mathbf{P}^{-1}_{L-1} \mathbf{M}^{-1}_{L-1}$, i.e. $h^L_{\boldsymbol{\theta}}$ and $h^L_{\boldsymbol{\eta}} \circ \mathbf{P}^{-1}_{L-1} \mathbf{M}^{-1}_{L-1}$ are identical linear functions supported on the full-dimensional domain $\mathcal{Z}^{L-1}_{\boldsymbol{\theta}}$. It follows that $\mathbf{W}_L = \mathbf{W}'_L \mathbf{P}^{-1}_{L-1} \mathbf{M}^{-1}_{L-1}$ and $\mathbf{b}_L = \mathbf{b}'_L$. $\qquad \square$

**Discussion of assumptions.**  Most of the theorem's assumptions have their origin in Lemma 4. The reason we restrict the domain of $h^{l:L}$ to the interior of $\mathcal{Z}^{l-1}$ is that we want $h^{l:L}$ to be defined on an open set (otherwise fold-sets become unwieldy). For similar reasons, we study only architectures with non-increasing widths; otherwise $\text{int } \mathcal{Z}^{l-1}$ may be empty. We conjecture that the theorem does *not* hold for more general architectures. If it does, the proof will likely go beyond fold-sets.

To guarantee transparency, our construction is such that for each input $\mathbf{z} \in \mathcal{Z}$ and layer $l \in [L-1]$, either $h^{1:l}_1(\mathbf{z}) > 0$ or $h^{1:l}_2(\mathbf{z}) > 0$. Transparency could in principle be achieved with just a single unit, but it would have to be positive everywhere. This is why we impose $d_l \geq 2$. Guaranteeing transparency for the first layer (whose inputs are not constrained to the positive quadrant) also necessitates boundedness of $\mathcal{Z}$. Boundedness can be lifted if we consider a slightly modified definition of transparency; proofs become more complicated though and we do not consider this crucial.

Almost all of the proof carries over to the case of leaky ReLU activations (where $\sigma$ is defined as $\sigma(\mathbf{u})_i = \max\{\alpha u_i, u_i\}$ for some small $\alpha > 0$). The part that does *not* carry over is our proof that $\mathbf{M}_l$ has only positive entries on the diagonal: In this part, we compare the slope of $h^{l:L}_{\boldsymbol{\theta}}$ for inputs on the positive and negative side of a given ReLU unit, and notice that the negative-side slope is 'singular' in the sense that some basis directions have zero magnitude. This particular argument does not work for the leaky ReLU, though we cannot rule out that a simple workaround exists.

## 7  DISCUSSION & FUTURE WORK

In this work, we have shown that for architectures with non-increasing widths, certain ReLU networks are almost uniquely identified by the function they implement. The result suggests that the function-equivalence classes of ReLU networks are surprisingly small, i.e. there may be only little redundancy in the way ReLU networks are parameterised, contrary to what is commonly believed.

This apparent contradiction could be explained in a number of ways:

- It could be the case that even though exact equivalence classes are small, approximate equivalence is much easier to achieve. That is, it could be that $\|h_{\boldsymbol{\theta}} - h_{\boldsymbol{\eta}}\| \leq \epsilon$ is satisfied

---

[6]See Appendix, Lemma A.19.
[7]See Appendix, Theorem A.1.

by a disproportionately larger class of parameters $\boldsymbol{\eta}$ than $\|h_{\boldsymbol{\theta}} - h_{\boldsymbol{\eta}}\| = 0$. This issue is related to the so-called inverse stability of the realisation map of neural nets, which is not yet well understood.

- Another possibility is that the kind of networks we consider in this paper is not representative of networks typically encountered in practice, i.e. it could be that 'typical networks' do not have well connected dependency graphs, and are therefore not easily identifiable.

- Finally, we have considered only architectures with non-increasing widths, whereas some previous theoretical work has assumed much wider intermediate layers compared to the input dimension. It is possible that parameterisation redundancy is much larger in such a regime compared to ours. However, gains from over-parameterisation have also been observed in practical settings with architectures not unlike those considered here.

We consider these questions important directions for further research. We also hypothesise that our analysis could be extended to convolutional and recurrent networks, and to other piece-wise linear activation functions such as leaky ReLU.

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

# A    APPENDIX

Use the following look-up table to find a particular lemma or theorem and its proof.

| | | |
|---|---|---|
| Lemma 1 | $\rightarrow$ | Lemma A.15 |
| Lemma 2 | $\rightarrow$ | Corollary A.1 |
| Lemma 3 | $\rightarrow$ | Lemma A.18 |
| Lemma 4 | $\rightarrow$ | Lemma A.17 |
| Theorem 1 | $\rightarrow$ | Theorem A.1 |

## A.1    PIECE-WISE LINEAR SURFACES

**Definition A.1** (Partition). *Let $\mathcal{S} \subseteq \mathcal{Z}$. We define the* partition of $\mathcal{Z}$ induced by $\mathcal{S}$, *denoted $\mathcal{P}_{\mathcal{Z}}(\mathcal{S})$, as the set of connected components of $\mathcal{Z} \setminus \mathcal{S}$.*

**Definition A.2** (Piece-wise hyperplane). *Let $\mathcal{P}$ be a partition of $\mathcal{Z}$. We say $\mathcal{H} \subseteq \mathcal{Z}$ is a* piece-wise hyperplane *with respect to partition $\mathcal{P}$, if $\mathcal{H} \neq \emptyset$ and there exist $(\mathbf{w}, b) \neq (\mathbf{0}, 0)$ and $P \in \mathcal{P}$ such that $\mathcal{H} = \{\mathbf{z} \in P \mid \mathbf{w}^{\mathsf{T}}\mathbf{z} + b = 0\}$.*

**Definition A.3** (Piece-wise linear surface / pwl. surface). *A set $\mathcal{S} \subseteq \mathcal{Z}$ is called a* piece-wise linear surface *on $\mathcal{Z}$ of order $\kappa$ if it can be written as $\mathcal{S} = \bigcup_{l \in [\kappa], i \in [n_l]} \mathcal{H}_i^l$, where each $\mathcal{H}_i^l$ is a piece-wise hyperplane with respect to $\mathcal{P}_{\mathcal{Z}}(\bigcup_{k \in [l-1], j \in [n_k]} \mathcal{H}_j^k)$, and no number smaller than $\kappa$ admits such a representation.*

**Lemma A.1.** *If $\mathcal{S}_1, \mathcal{S}_2$ are piece-wise linear surfaces on $\mathcal{Z}$ of order $k_1$ and $k_2$, then $\mathcal{S}_1 \cup \mathcal{S}_2$ is a piece-wise linear surface on $\mathcal{Z}$ of order at most $\max\{k_1, k_2\}$.*

*Proof.* Let $\mathcal{S}_1 = \bigcup_{l \in [k_1], i \in [n_l]} \mathcal{H}_i^l$ and $\mathcal{S}_2 = \bigcup_{l \in [k_2], i \in [m_l]} \mathcal{G}_i^l$ be the pwl. surface representations of $\mathcal{S}_1, \mathcal{S}_2$. Given $\mathcal{H}_i^l$, consider the partition

$$\mathcal{P} := \mathcal{P}_{\mathcal{Z}}\left( \bigcup_{k \in [l-1], j \in [n_k]} \mathcal{H}_j^k \cup \bigcup_{k \in [\max\{l-1, k_2\}], j \in [m_k]} \mathcal{G}_j^k \right). \tag{1}$$

We can write $\mathcal{H}_i^l = \bigcup_{P \in \mathcal{P}} \mathcal{H}_i^l \cap P$ and denote the nonempty intersections $\mathcal{H}_i^l \cap P$ as $\{\bar{\mathcal{H}}_{i,j}^l\}_j$. Similarly, we decompose $\mathcal{G}_i^l = \bigcup_j \bar{\mathcal{G}}_{i,j}^l$. Then $\mathcal{S}_1 \cup \mathcal{S}_2 = \bigcup_{l \in [\max\{k_1, k_2\}]}(\bigcup_{i,j} \bar{\mathcal{H}}_{i,j}^l \cup \bigcup_{i,j} \bar{\mathcal{G}}_{i,j}^l)$, where each $\bar{\mathcal{H}}_{i,j}^l$ and $\bar{\mathcal{G}}_{i,j}^l$ is a piece-wise hyperplane wrt. $\mathcal{P} = \mathcal{P}_{\mathcal{Z}}(\bigcup_{k \in [l-1]}(\bigcup_{i',j'} \bar{\mathcal{H}}_{i',j'}^k \cup \bigcup_{i',j'} \bar{\mathcal{G}}_{i',j'}^k))$. $\qquad\square$

Given sets $\mathcal{Z}$ and $\mathcal{S} \subseteq \mathcal{Z}$, we introduce the notation

$$\square_i \mathcal{S} := \bigcup \{\mathcal{S}' \subseteq \mathcal{S} \mid \mathcal{S}' \text{ is a pwl. surface on } \mathcal{Z} \text{ of order at most } i\}. \tag{2}$$

(The dependence on $\mathcal{Z}$ is suppressed.) By Lemma A.1, $\square_i \mathcal{S}$ is itself a pwl. surface on $\mathcal{Z}$ of order at most $i$.

**Lemma A.2.** *For $i \leq j$ and any set $\mathcal{S}$, we have $\square_i \square_j \mathcal{S} = \square_j \square_i \mathcal{S} = \square_i \mathcal{S}$.*

*Proof.* We will need these definitions:

$$\square_i \mathcal{S} = \bigcup \{\mathcal{S}'' \subseteq \mathcal{S} \mid \mathcal{S}'' \text{ is a pwl. surface of order at most } i\}, \tag{3}$$

$$\square_j \mathcal{S} = \bigcup \{\mathcal{S}'' \subseteq \mathcal{S} \mid \mathcal{S}'' \text{ is a pwl. surface of order at most } j\}, \tag{4}$$

$$\square_i \square_j \mathcal{S} = \bigcup \{\mathcal{S}'' \subseteq \square_j \mathcal{S} \mid \mathcal{S}'' \text{ is a pwl. surface of order at most } i\}, \tag{5}$$

$$\square_j \square_i \mathcal{S} = \bigcup \{\mathcal{S}'' \subseteq \square_i \mathcal{S} \mid \mathcal{S}'' \text{ is a pwl. surface of order at most } j\}. \tag{6}$$

Consider first the equality $\square_j \square_i \mathcal{S} = \square_i \mathcal{S}$. We know that $\square_j \square_i \mathcal{S} \subseteq \square_i \mathcal{S}$ because the square operator always yields a subset. At the same time, $\square_i \mathcal{S} \subseteq \square_j \square_i \mathcal{S}$, because $\square_i \mathcal{S}$ satisfies the condition for membership in (6).

To prove the equality $\square_i\square_j\mathcal{S} = \square_i\mathcal{S}$, we use the inclusion $\square_j\mathcal{S} \subseteq \mathcal{S}$ to deduce $\square_i\square_j\mathcal{S} \subseteq \square_i\mathcal{S}$. Now let $\mathcal{S}'' \subseteq \mathcal{S}$ be one of the sets under the union in (3), i.e. it is a pwl. surface of order at most $i$. Then it is also a pwl. surface of order at most $j$, implying $\mathcal{S}'' \subseteq \square_j\mathcal{S}$. This means $\mathcal{S}''$ is also one of the sets under the union in (5), proving that $\square_i\mathcal{S} \subseteq \square_i\square_j\mathcal{S}$. $\qquad\square$

**Lemma A.3.** *Let $\mathcal{Z}$ and $\mathcal{S} \subseteq \mathcal{Z}$ be sets. Then one can write $\square_{k+1}\mathcal{S} = \square_k\mathcal{S} \cup \bigcup_i \mathcal{H}_i$ where $\mathcal{H}_i$ are piece-wise hyperplanes wrt. $\mathcal{P}_\mathcal{Z}(\square_k\mathcal{S})$.*

*Proof.* Let $\square_{k+1}\mathcal{S} = \bigcup_{l\in[\kappa],i\in[n_l]} \mathcal{H}_i^l$ be the pwl. surface representation of $\square_{k+1}\mathcal{S}$. If $\kappa \leq k$, then $\square_{k+1}\mathcal{S} = \square_k\mathcal{S}$ and we are done. Otherwise, $\bigcup_{l\in[k],i\in[n_k]} \mathcal{H}_i^l \subseteq \square_k\mathcal{S}$, implying

$$\square_{k+1}\mathcal{S} \subseteq \square_k\mathcal{S} \cup \bigcup_{i\in[n_{k+1}]} \mathcal{H}_i^{k+1}. \tag{7}$$

At the same time, $\square_k\mathcal{S} \cup \bigcup_{i\in[n_{k+1}]} \mathcal{H}_i^{k+1}$ is a pwl. surface of order at most $k+1$ because $\square_k\mathcal{S}$ is a pwl. surface of order at most $k$ and $\mathcal{H}_i^{k+1}$ can be decomposed into piece-wise hyperplanes wrt. $\mathcal{P}_\mathcal{Z}(\square_k\mathcal{S})$. Therefore, $\square_k\mathcal{S} \cup \bigcup_{i\in[n_{k+1}]} \mathcal{H}_i^{k+1} \subseteq \square_{k+1}\mathcal{S}$, implying in fact equality. $\qquad\square$

**Definition A.4** (Canonical representation of a pwl. surface)*. Let $\mathcal{S}$ be a pwl. surface on $\mathcal{Z}$. The pwl. surface representation $\mathcal{S} = \bigcup_{l\in[\kappa],i\in[n_l]} \mathcal{H}_i^l$ is called* canonical *if $\bigcup_{l\in[k],i\in[n_l]} \mathcal{H}_i^l = \square_k\mathcal{S}$ for all $k \in [\kappa]$, and each $\mathcal{H}_i^l$ is distinct.*

**Lemma A.4.** *If $\mathcal{S} = \bigcup_{l\in[\kappa],i\in[n_l]} \mathcal{H}_i^l$ is a pwl. surface in canonical form, then $\kappa$ is the order of $\mathcal{S}$.*

*Proof.* Denote the order of $\mathcal{S}$ by $\lambda$. By the definition of order, $\lambda \leq \kappa$, and $\mathcal{S} = \square_\lambda\mathcal{S}$. Then, since $\mathcal{S} = \bigcup_{l\in[\kappa],i\in[n_l]} \mathcal{H}_i^l$ is a canonical representation, we have

$$\bigcup_{l\in[\lambda],i\in[n_l]} \mathcal{H}_i^l = \square_\lambda\mathcal{S} = \mathcal{S} = \bigcup_{l\in[\kappa],i\in[n_l]} \mathcal{H}_i^l. \tag{8}$$

It follows that $\kappa = \lambda$. $\qquad\square$

**Lemma A.5.** *Every pwl. surface has a canonical representation.*

*Proof.* The inclusion $\bigcup_{l\in[k],i\in[n_l]} \mathcal{H}_i^l \subseteq \square_k\mathcal{S}$ holds for any representation. We will show the other inclusion by induction in the order of $\mathcal{S}$. If $\mathcal{S}$ is order one, $\square_1\mathcal{S} \subseteq \mathcal{S} = \bigcup_{i\in[n_1]} \mathcal{H}_i^1$ holds for any representation and we are done. Now assume the lemma holds up to order $\kappa - 1$, and let $\mathcal{S}$ be order $\kappa$. Then by Lemma A.3, $\mathcal{S} = \square_\kappa\mathcal{S} = \square_{\kappa-1}\mathcal{S} \cup \bigcup_i \mathcal{H}_i^\kappa$, where $\mathcal{H}_i^\kappa$ are piece-wise hyperplanes wrt. $\mathcal{P}_\mathcal{Z}(\square_{\kappa-1}\mathcal{S})$. By the inductive assumption, $\square_{\kappa-1}\mathcal{S}$ has a canonical representation, say $\square_{\kappa-1}\mathcal{S} = \bigcup_{l\in[\kappa-1],i\in[n_l]} \mathcal{H}_i^l$. We claim that $\mathcal{S} = \bigcup_{l\in[\kappa],i\in[n_l]} \mathcal{H}_i^l$ is a canonical representation of $\mathcal{S}$. If $k = \kappa$, then clearly $\square_k\mathcal{S} \subseteq \mathcal{S} = \bigcup_{l\in[\kappa],i\in[n_l]} \mathcal{H}_i^l$. If $k \in [\kappa - 1]$, then by Lemma A.2, $\square_k\mathcal{S} = \square_k\square_{\kappa-1}\mathcal{S} = \bigcup_{l\in[k],i\in[n_l]} \mathcal{H}_i^l$, where we have used the canonical representation of $\square_{\kappa-1}\mathcal{S}$.

Finally, distinctness of $\mathcal{H}_i^l$ can be ensured by throwing away duplicates. $\qquad\square$

**Lemma A.6.** *Let $\mathcal{Z}$ be an open set. If $\mathcal{S}$ is a piece-wise linear surface on $\mathcal{Z}$, and if $\mathcal{S} = \bigcup_{l\in[\kappa],i\in[n_l]} \mathcal{H}_i^l$ and $\mathcal{S} = \bigcup_{k\in[\kappa],j\in[m_k]} \mathcal{G}_j^k$ are two canonical representations of $\mathcal{S}$, then for all $l \in [\kappa]$, $n_l = m_l$ and there exists a permutation $\pi : [n_l] \to [n_l]$ such that $\mathcal{H}_i^l = \mathcal{G}_{\pi(i)}^l$. In other words, the canonical representation is unique up to within-order indexing.*

*Proof.* Let $k \in [\kappa]$. Because both representations are canonical, we have

$$\square_{k-1}\mathcal{S} \cup \bigcup_{i\in[n_k]} \mathcal{H}_i^k = \square_k\mathcal{S} = \square_{k-1}\mathcal{S} \cup \bigcup_{j\in[m_k]} \mathcal{G}_j^k, \tag{9}$$

where $\mathcal{H}_i^k$ and $\mathcal{G}_j^k$ are piece-wise hyperplanes wrt. $\mathcal{P}_\mathcal{Z}(\square_{k-1}\mathcal{S})$. Then for each $P \in \mathcal{P}_\mathcal{Z}(\square_{k-1}\mathcal{S})$,

$$P \cap \bigcup_{i\in[n_k]} \mathcal{H}_i^k = P \cap \bigcup_{j\in[m_k]} \mathcal{G}_j^k, \tag{10}$$

where on both sides above we have a union of hyperplanes on an open set. The claim follows. $\qquad\square$

**Definition A.5** (Dependency graph of a pwl. surface). *Let $\mathcal{S}$ be a piece-wise linear surface on $\mathcal{Z}$, and let $\mathcal{S} = \bigcup_{l \in [\kappa], i \in [n_l]} \mathcal{H}_i^l$ be its canonical representation. We define the* dependency graph of *$\mathcal{S}$ as the directed graph that has the piece-wise hyperplanes $\{\mathcal{H}_i^l\}_{l,i}$ as vertices, and has an edge $\mathcal{H}_i^l \to \mathcal{H}_j^k$ iff $l < k$ and $\mathcal{H}_i^l \cap \operatorname{cl} \mathcal{H}_j^k \neq \emptyset$.*

## A.2 ReLU NETWORKS AND FOLDS

We denote by $\sigma$ the ReLU function: $\sigma(\mathbf{u})_i = \max\{0, u_i\}$ for $i \in [\dim(\mathbf{u})]$.

**Definition A.6** (ReLU network). *Let $\mathcal{Z} \subseteq \mathbb{R}^{d_0}$ with $d_0 \geq 2$ be a nonempty open set, and let $\boldsymbol{\theta} \triangleq (\mathbf{W}_1, \mathbf{b}_1, \dots, \mathbf{W}_L, b_L)$ be the network's parameters, with $\mathbf{W}_l \in \mathbb{R}^{d_l \times d_{l-1}}$, $\mathbf{b}_l \in \mathbb{R}^{d_l}$, and $d_L = 1$. A ReLU network parameterised by $\boldsymbol{\theta}$ is the function $h_{\boldsymbol{\theta}} : \mathcal{Z} \to \mathbb{R}$, defined by*

$$h_{\boldsymbol{\theta}} \triangleq h_{\boldsymbol{\theta}}^L \circ \sigma \circ h_{\boldsymbol{\theta}}^{L-1} \circ \cdots \circ \sigma \circ h_{\boldsymbol{\theta}}^1, \tag{11}$$

*where $h_{\boldsymbol{\theta}}^l(\mathbf{z}) = \mathbf{W}_l \cdot \mathbf{z} + \mathbf{b}_l$. For $1 \leq l \leq k \leq L$, we also denote*

$$h_{\boldsymbol{\theta}}^{l:k} \triangleq h_{\boldsymbol{\theta}}^k \circ \sigma \circ h_{\boldsymbol{\theta}}^{k-1} \cdots \circ \sigma \circ h_{\boldsymbol{\theta}}^l, \tag{12}$$

$$\check{h}_{\boldsymbol{\theta}}^{l:k} \triangleq \sigma \circ h_{\boldsymbol{\theta}}^{l:k}. \tag{13}$$

For a ReLU network $h_{\boldsymbol{\theta}} : \mathcal{Z} \to \mathbb{R}$ and $l \in [L-1]$, denote $\mathcal{Z}_{\boldsymbol{\theta}}^l \triangleq \{\check{h}_{\boldsymbol{\theta}}^{1:l}(\mathbf{z}) \mid \mathbf{z} \in \mathcal{Z}\}$. Also, for convenience, define $\mathcal{Z}_{\boldsymbol{\theta}}^0 \triangleq \mathcal{Z}$. (We will omit the subscript $\boldsymbol{\theta}$ when it is clear from the context.) We write $f|_{\mathcal{A}}$ to denote the restriction of the function $f$ to the domain $\mathcal{A}$.

**Definition A.7** (Activation indicator). *A tuple $\mathbf{I} \triangleq (\mathbf{I}_1, \dots, \mathbf{I}_{L-1})$ is called an* activation indicator *if $\mathbf{I}_l = \operatorname{diag}(\mathbf{i}_l) \in \mathbb{R}^{d_l \times d_l}$ and $\mathbf{i}_l \in \{0, 1\}^{d_l}$ for $l \in [L-1]$. It is called* non-trivial *if $\mathbf{i}_l \neq \mathbf{0}$ for all $l \in [L-1]$ and* non-trivial up to $k$ *if $\mathbf{i}_l \neq \mathbf{0}$ for all $l \in [k]$.*

Given a parameter vector $\boldsymbol{\theta} \triangleq (\mathbf{W}_1, \mathbf{b}_1, \dots, \mathbf{W}_L, \mathbf{b}_L)$ and an activation indicator $\mathbf{I}$, we introduce the notation

$$\mathbf{w}_i^l(\boldsymbol{\theta}, \mathbf{I}) \triangleq \mathbf{e}_i^{\intercal} \mathbf{W}_l \mathbf{I}_{l-1} \mathbf{W}_{l-1} \cdots \mathbf{I}_1 \mathbf{W}_1, \tag{14}$$

$$b_i^l(\boldsymbol{\theta}, \mathbf{I}) \triangleq \mathbf{e}_i^{\intercal} \sum_{k=1}^{l} \mathbf{W}_l \mathbf{I}_{l-1} \cdots \mathbf{W}_{k+1} \mathbf{I}_k \mathbf{b}_k. \tag{15}$$

(We will omit the argument $\boldsymbol{\theta}$ when it is clear from the context.) These quantities characterise the different linear pieces implemented by the network's units. Also define $\mathbf{I}^{\boldsymbol{\theta}}(\mathbf{z}) \triangleq (\mathbf{I}_1^{\boldsymbol{\theta}}(\mathbf{z}), \dots, \mathbf{I}_{L-1}^{\boldsymbol{\theta}}(\mathbf{z}))$ as the activation indicator for a specific input: $I_l^{\boldsymbol{\theta}}(\mathbf{z})[i, i] \triangleq \mathbb{1}\{h_i^{1:l}(\mathbf{z}) \geq 0\}$ for all $(l, i)$.

**Lemma A.7.** *In a ReLU network with parameters $\boldsymbol{\theta} = (\mathbf{W}_1, \mathbf{b}_1, \dots, \mathbf{W}_L, b_L)$, the pre-activations satisfy $h_i^{1:l}(\mathbf{z}) \in \{\mathbf{w}_i^l(\boldsymbol{\theta}, \mathbf{I}) \cdot \mathbf{z} + b_i^l(\boldsymbol{\theta}, \mathbf{I})\}_{\mathbf{I}}$, where the indexing runs over all possible activation indicators $\mathbf{I}$. More precisely, $h_i^{1:l}(\mathbf{z}) = \mathbf{w}_i^l(\boldsymbol{\theta}, \mathbf{I}^{\boldsymbol{\theta}}(\mathbf{z})) \cdot \mathbf{z} + b_i^l(\boldsymbol{\theta}, \mathbf{I}^{\boldsymbol{\theta}}(\mathbf{z}))$.*

*Proof.* Left as exercise. $\square$

**Definition A.8** (Fold-set). *Let $\mathcal{Z}$ be an open set, and $f : \mathcal{Z} \to \mathbb{R}$ a continuous, piece-wise linear function. We define the* fold-set *of $f$, denoted by $\mathcal{F}(f)$, as the set of all points at which $f$ is non-differentiable.*

**Definition A.9** (Positive / negative in a neighbourhood). *Let $\mathcal{Z}$ be an open set. The function $f : \mathcal{Z} \to \mathbb{R}$ is* positive (negative) in the neighbourhood of $\mathbf{z} \in \mathcal{Z}$ *if for any $\epsilon > 0$ there exists $\mathbf{z}' \in \mathcal{B}_\epsilon(\mathbf{z})$ such that $f(\mathbf{z}') > 0$ ($f(\mathbf{z}') < 0$).*

**Definition A.10** (Unit fold-set). *Let $h_{\boldsymbol{\theta}} : \mathcal{Z} \to \mathbb{R}$ be a ReLU network. We define the* unit $(l, i)$ fold-set *of $h_{\boldsymbol{\theta}}$, denoted $\mathcal{F}_i^l(h_{\boldsymbol{\theta}})$, as the set of all $\mathbf{z} \in \mathcal{Z}$ where $h_{\boldsymbol{\theta}}^{1:l}[i](\mathbf{z}) = 0$ and $h_{\boldsymbol{\theta}}^{1:l}[i]$ is positive in the neighbourhood of $\mathbf{z}$.*

**Lemma A.8.** *Let $\mathcal{Z}$ be an open set, and $f : \mathcal{Z} \to \mathbb{R}$ a continuous piece-wise linear function. Then $\mathcal{F}(\sigma \circ f)$ consists of those $\mathbf{z} \in \mathcal{Z}$ that satisfy*

- *$f(\mathbf{z}) > 0$ and $\mathbf{z} \in \mathcal{F}(f)$, or*

- $f(\mathbf{z}) = 0$ *and $f$ is positive in the neighbourhood of* $\mathbf{z}$.

*Proof.* We will prove that if $\mathbf{z}$ satisfies any of the two conditions, then $\mathbf{z} \in \mathcal{F}(\sigma \circ f)$, and if it violates both, then $\mathbf{z} \in \mathcal{F}(\sigma \circ f)^c$. We begin with the latter implication.

Let $\mathbf{z}$ be such that $f(\mathbf{z}) > 0$ and $\mathbf{z} \notin \mathcal{F}(f)$, i.e. $f$ is differentiable at $\mathbf{z}$. Since $f$ is piece-wise linear, there exists $\epsilon > 0$ such that all of $\mathcal{B}_\epsilon(\mathbf{z})$ lies inside a single linear region of $f$ and $f(\mathcal{B}_\epsilon(\mathbf{z})) \subseteq (0, \infty]$. Then, on $\mathcal{B}_\epsilon(\mathbf{z})$, the ReLU behaves like an identity, implying $\sigma \circ f$ is differentiable at $\mathbf{z}$, proving that $\mathbf{z} \in \mathcal{F}(\sigma \circ f)^c$. Next, consider $\mathbf{z}$ such that $f(\mathbf{z}) = 0$. For it to violate the second condition, there must exist a ball $\mathcal{B}_\epsilon(\mathbf{z})$ around $\mathbf{z}$ such that $f(\mathcal{B}_\epsilon(\mathbf{z})) \subseteq (-\infty, 0]$. (This is also true if $f(\mathbf{z}) < 0$.) Then, on $\mathcal{B}_\epsilon(\mathbf{z})$, the ReLU behaves like a constant zero, implying that $\sigma \circ f$ is differentiable at $\mathbf{z}$.

We now prove the other implication. If $f(\mathbf{z}) > 0$ and $\mathbf{z} \in \mathcal{F}(f)$, then there exists $\epsilon > 0$ such that $f(\mathcal{B}_\epsilon(\mathbf{z})) \subseteq (0, \infty]$, which guarantees that the ReLU behaves like an identity on $\mathcal{B}_\epsilon(\mathbf{z})$. In this ball, we have $\sigma \circ f = f$, so $\mathbf{z} \in \mathcal{F}(\sigma \circ f)$.

If $f(\mathbf{z}) = 0$ and $f$ is positive in the neighbourhood of $\mathbf{z}$, we distinguish several cases. If $\mathbf{z} \notin \mathcal{F}(f)$, then there exists a ball $\mathcal{B}_\delta(\mathbf{z})$ on which $f$ behaves linearly, i.e. $\sigma(f(\mathbf{z})) = \sigma(\mathbf{w}^\intercal \mathbf{z} + b)$, implying $\mathbf{z} \in \mathcal{F}(\sigma \circ f)$. If $\mathbf{z} \in \mathcal{F}(f)$ and, in addition, there exists a ball $\mathcal{B}_\delta(\mathbf{z})$ such that $f(\mathcal{B}_\delta(\mathbf{z})) \subseteq [0, \infty)$, then the ReLU behaves like an identity on $\mathcal{B}_\delta(\mathbf{z})$ and $\mathbf{z} \in \mathcal{F}(\sigma \circ f)$. The final case is $\mathbf{z} \in \mathcal{F}(f)$ such that $f$ attains both positive and negative values in its neighbourhood. Since $f$ is piece-wise linear, there exist $\mathbf{p}, \mathbf{n}$ such that $f(\mathbf{z} + \epsilon \mathbf{n}) < 0 < f(\mathbf{z} + \epsilon \mathbf{p})$, and $\mathbf{z} + \epsilon \mathbf{p}, \mathbf{z} + \epsilon \mathbf{n} \notin \mathcal{F}(f)$ for all $\epsilon \in (0, 1]$. Then $\nabla(\sigma \circ f)(\mathbf{z} + \epsilon \mathbf{p}) \neq \mathbf{0}$ and $\nabla(\sigma \circ f)(\mathbf{z} + \epsilon \mathbf{n}) = \mathbf{0}$, yielding $\mathbf{z} \in \mathcal{F}(\sigma \circ f)$. $\square$

**Lemma A.9.** *Let $\mathcal{Z}$ be an open set, and let $f_1, \dots, f_n : \mathcal{Z} \to \mathbb{R}$ be continuous, piece-wise linear functions. For any $w_1, \dots, w_n \in \mathbb{R}$, define $f = \sum_{i=1}^n w_i f_i$. Then $\mathcal{F}(f) \subseteq \bigcup_{i=1}^n \mathcal{F}(f_i)$.*

*Proof.* Left as exercise. $\square$

### A.3 GENERAL AND TRANSPARENT RELU NETWORKS

**Lemma A.10.** *For all $\boldsymbol{\theta}$ except a closed zero-measure set,*

$$\mathrm{rank}(\mathbf{W}_l \mathbf{I}_{l-1} \cdots \mathbf{I}_k \mathbf{W}_k) = \min \{d_{k-1}, \mathrm{rank}(\mathbf{I}_k), \dots, \mathrm{rank}(\mathbf{I}_{l-1}), d_l\}, \tag{16}$$

$$\mathrm{rank}(\mathbf{I}_l \mathbf{W}_l \mathbf{I}_{l-1} \cdots \mathbf{I}_k \mathbf{W}_k) = \min \{d_{k-1}, \mathrm{rank}(\mathbf{I}_k), \dots, \mathrm{rank}(\mathbf{I}_{l-1}), \mathrm{rank}(\mathbf{I}_l)\}, \tag{17}$$

*for all activation indicators $\mathbf{I}$ and all $k \leq l$.*

*Proof.* First, notice that (16) is just a special case of (17) with $\mathbf{I}_l$ equal to the identity matrix. It therefore suffices to prove (17).

To further simplify, we will prove the statement for a single fixed activation indicator $\mathbf{I}$. Then if $\Theta(\mathbf{I})$ is the set of networks for which (17) holds given $\mathbf{I}$, and $\Theta(\mathbf{I})$ contains all networks except a closed zero-measure set, then also $\bigcap_{\mathbf{I}} \Theta(\mathbf{I})$ contains all networks except a closed zero-measure set, proving the lemma.

Let us hence fix $\mathbf{I}$, and let $k \in [L]$. We proceed by induction. For the initial step, notice that the matrix $\mathbf{I}_k \mathbf{W}_k$ is just $\mathbf{W}_k$ with some rows replaced by zeroes. The rank of such a matrix is the same as the matrix obtained by removing the zero rows, which has size $(\mathrm{rank}(\mathbf{I}_k), d_{k-1})$. For all $\mathbf{W}_k$ except a closed zero-measure set, this matrix has rank $\min \{d_{k-1}, \mathrm{rank}(\mathbf{I}_k)\}$.

For the inductive step, denote $\bar{\mathbf{W}}_i := \mathbf{I}_i \mathbf{W}_i \cdots \mathbf{I}_k \mathbf{W}_k$ and

$$r_i := \min \{d_{k-1}, \mathrm{rank}(\mathbf{I}_k), \dots, \mathrm{rank}(\mathbf{I}_i)\}. \tag{18}$$

We assume that $\mathrm{rank}(\bar{\mathbf{W}}_{i-1}) = r_{i-1}$ and want to prove the same for $i$. Notice that for all $\mathbf{W}_i$ except a closed zero-measure set, any $r_i$ rows of $\mathbf{W}_i$ are linearly independent and their span intersects with $\ker(\bar{\mathbf{W}}_{i-1}^\intercal)$ only at $\mathbf{0}$. To see this, recall that by the inductive assumption, $\mathrm{rank}(\bar{\mathbf{W}}_{i-1}^\intercal) = r_{i-1}$, so $\ker(\bar{\mathbf{W}}_{i-1}^\intercal)$ has dimension $d_{i-1} - r_{i-1}$. We can concatenate any $r_i$-subset of rows of $\mathbf{W}_i$ to the basis of $\ker(\bar{\mathbf{W}}_{i-1}^\intercal)$ to obtain a matrix of size $(r_i + d_{i-1} - r_{i-1}, d_{i-1})$, which is a wide matrix, because $r_i \leq r_{i-1}$. Hence, its rows are linearly independent for all $\mathbf{W}_i$ except a closed zero-measure set.

We now prove that $\mathrm{rank}(\mathbf{I}_i \mathbf{W}_i \bar{\mathbf{W}}_{i-1}) = \min \left\{ \mathrm{rank}(\bar{\mathbf{W}}_{i-1}), \mathrm{rank}(\mathbf{I}_i) \right\} \triangleq r_i$. The "$\leq$" direction is immediate. For the "$\geq$" direction, we distinguish between two cases. If $\mathrm{rank}(\mathbf{I}_i) \leq \mathrm{rank}(\bar{\mathbf{W}}_{i-1})$, let $\mathbf{v}_1, \ldots \mathbf{v}_{r_i}$ be the (linearly independent) nonzero rows of $\mathbf{I}_i \mathbf{W}_i$. We want to show that $\left\{ \mathbf{v}_j^\mathsf{T} \bar{\mathbf{W}}_{i-1} \right\}_j$ are linearly independent, i.e. that $\mathbf{I}_i \mathbf{W}_i \bar{\mathbf{W}}_{i-1}$ has at least $r_i$ linearly independent rows. If $\sum_{j=1}^{r_i} \lambda_j \mathbf{v}_j^\mathsf{T} \bar{\mathbf{W}}_{i-1} = \mathbf{0}$, then $\sum_{j=1}^{r_i} \lambda_j \mathbf{v}_j \in \ker(\bar{\mathbf{W}}_{i-1}^\mathsf{T})$, which by assumption implies $\sum \lambda_j \mathbf{v}_j = \mathbf{0}$. By the independence of $\{\mathbf{v}_j\}$, we obtain $\lambda_j = 0$, i.e. $\left\{ \mathbf{v}_j^\mathsf{T} \bar{\mathbf{W}}_{i-1} \right\}_j$ are linearly independent, and $\mathrm{rank}(\mathbf{I}_i \mathbf{W}_i \bar{\mathbf{W}}_{i-1}) = r_i$.

If $\mathrm{rank}(\mathbf{I}_i) > \mathrm{rank}(\bar{\mathbf{W}}_{i-1})$, we can reduce the problem to the case $\mathrm{rank}(\mathbf{I}_i) \leq \mathrm{rank}(\bar{\mathbf{W}}_{i-1})$ by observing that $\mathrm{rank}(\mathbf{I}_i \mathbf{W}_i \bar{\mathbf{W}}_{i-1}) \geq \mathrm{rank}(\mathbf{J}_i \mathbf{W}_i \bar{\mathbf{W}}_{i-1})$ if $\mathbf{J}_i$ equals $\mathbf{I}_i$ only with some 1's replaced by 0's. We can thus take any such $\mathbf{J}_i$ and apply the argument from the previous paragraph to obtain $\mathrm{rank}(\mathbf{I}_i \mathbf{W}_i \bar{\mathbf{W}}_{i-1}) \geq \mathrm{rank}(\mathbf{J}_i \mathbf{W}_i \bar{\mathbf{W}}_{i-1}) \geq r_i$. $\qquad \square$

**Lemma A.11.** *For all $\boldsymbol{\theta}$ except a closed zero-measure set, the following holds. Let $(l, i)$, $(k, j)$ be any units, let $\mathbf{I}$ be an activation indicator non-trivial up to $l-1$, and let $\mathbf{J}$ be an activation indicator non-trivial up to $k-1$, such that $(l, i, \mathbf{I}_{1:l-1}) \neq (k, j, \mathbf{J}_{1:k-1})$. Then, for all scalars $c \in \mathbb{R}$, it holds that $[\mathbf{w}_i^l(\boldsymbol{\theta}, \mathbf{I}), b_i^l(\boldsymbol{\theta}, \mathbf{I})] \neq c[\mathbf{w}_j^k(\boldsymbol{\theta}, \mathbf{J}), b_j^k(\boldsymbol{\theta}, \mathbf{J})]$.*

*Proof.* First, we exclude from consideration all $\boldsymbol{\theta} = (\mathbf{W}_1, \mathbf{b}_1, \ldots, \mathbf{W}_L, \mathbf{b}_L)$ such that $\mathbf{e}_i^\mathsf{T} \mathbf{W}_l \mathbf{I}_{l-1} \mathbf{W}_{l-1} \cdots \mathbf{I}_k \mathbf{W}_k \mathbf{e}_j = 0$ for some $l, k, i, j$, and some $\mathbf{I}$ non-trivial up to $l-1$. Since for any fixed $(l, k, i, j, \mathbf{I})$, the set of $\boldsymbol{\theta}$ satisfying the above is the set of roots of a non-trivial polynomial in $\boldsymbol{\theta}$, it is zero-measure and closed. Because there are only finitely many configurations of $(l, k, i, j, \mathbf{I})$, we have thus excluded a closed zero-measure set of parameters. We will denote its complement $\boldsymbol{\Theta}^*$.

From now on, we assume $\boldsymbol{\theta} \in \boldsymbol{\Theta}^*$. Notice that the case $c = 0$ of the lemma is thus automatically satisfied, since $\mathbf{w}_i^l(\boldsymbol{\theta}, \mathbf{I}) \triangleq \mathbf{e}_i^\mathsf{T} \mathbf{W}_l \mathbf{I}_{l-1} \mathbf{W}_{l-1} \cdots \mathbf{I}_1 \mathbf{W}_1 \neq \mathbf{0}$ by the definition of $\boldsymbol{\Theta}^*$. In the following, we can therefore assume $c \neq 0$ and treat $(l, i, \mathbf{I})$ and $(k, j, \mathbf{J})$ symmetrically.

Denote by $\boldsymbol{\Theta}^\neg \subseteq \boldsymbol{\Theta}^*$ the set of parameters $\boldsymbol{\theta}$ for which the lemma does *not* hold; we need to show that $\boldsymbol{\Theta}^\neg$ is closed and zero-measure. We start by showing the latter property by contradiction.

Suppose $\boldsymbol{\Theta}^\neg$ is positive-measure. We know that for all $\boldsymbol{\theta} \in \boldsymbol{\Theta}^\neg$, there exist triples $(l, i, \mathbf{I})$, $(k, j, \mathbf{J})$ as stated in the lemma, and a scalar $c \in \mathbb{R}$ such that $[\mathbf{w}_i^l(\boldsymbol{\theta}, \mathbf{I}), b_i^l(\boldsymbol{\theta}, \mathbf{I})] = c[\mathbf{w}_j^k(\boldsymbol{\theta}, \mathbf{J}), b_j^k(\boldsymbol{\theta}, \mathbf{J})]$. Let $\mathcal{C}$ denote the set of all triplet-pairs $((l, i, \mathbf{I}), (k, j, \mathbf{J}))$ satisfying the conditions of the lemma; then the previous statement can be written as

$$\boldsymbol{\Theta}^\neg \subseteq \bigcup_{((l,i,\mathbf{I}),(k,j,\mathbf{J})) \in \mathcal{C}} \left\{ \boldsymbol{\theta} \in \boldsymbol{\Theta}^* \,|\, \exists c \in \mathbb{R} : [\mathbf{w}_i^l(\boldsymbol{\theta}, \mathbf{I}), b_i^l(\boldsymbol{\theta}, \mathbf{I})] = c[\mathbf{w}_j^k(\boldsymbol{\theta}, \mathbf{J}), b_j^k(\boldsymbol{\theta}, \mathbf{J})] \right\}. \quad (19)$$

Since $\mathcal{C}$ is finite, there exist $((l, i, \mathbf{I}), (k, j, \mathbf{J})) \in \mathcal{C}$ for which the set under the union (call it $\boldsymbol{\Theta}'$) is positive-measure.

We now consider two cases. If $(l, i) \neq (k, j)$, then observe that $\boldsymbol{\Theta}'$ must contain some $\boldsymbol{\theta}, \boldsymbol{\theta}'$ such that $\boldsymbol{\theta} = (\mathbf{W}_1, \mathbf{b}_1, \ldots, \mathbf{W}_L, \mathbf{b}_L)$ and $\boldsymbol{\theta}' = (\mathbf{W}_1, \mathbf{b}_1, \ldots, \mathbf{W}_l, \mathbf{b}_l + \delta \mathbf{e}_i, \ldots, \mathbf{W}_L, \mathbf{b}_L)$, where $\delta \neq 0$ and $l \geq k$. By membership in $\boldsymbol{\Theta}'$, there exist $c, c' \in \mathbb{R}$ such that

$$[\mathbf{w}_i^l(\boldsymbol{\theta}, \mathbf{I}), b_i^l(\boldsymbol{\theta}, \mathbf{I})] = c[\mathbf{w}_j^k(\boldsymbol{\theta}, \mathbf{J}), b_j^k(\boldsymbol{\theta}, \mathbf{J})], \quad (20)$$

$$[\mathbf{w}_i^l(\boldsymbol{\theta}', \mathbf{I}), b_i^l(\boldsymbol{\theta}', \mathbf{I})] = c'[\mathbf{w}_j^k(\boldsymbol{\theta}', \mathbf{J}), b_j^k(\boldsymbol{\theta}', \mathbf{J})]. \quad (21)$$

Notice that $\mathbf{w}_i^l, \mathbf{w}_j^k$ do not depend on the $b_l[i]$-component of $\boldsymbol{\theta}$, and neither does $b_j^k$ because $k \leq l$ and $(k, j) \neq (l, i)$. It follows that $[\mathbf{w}_j^k(\boldsymbol{\theta}, \mathbf{J}), b_j^k(\boldsymbol{\theta}, \mathbf{J})] = [\mathbf{w}_j^k(\boldsymbol{\theta}', \mathbf{J}), b_j^k(\boldsymbol{\theta}', \mathbf{J})] =: \mathbf{v}$. Notice also that $[\mathbf{w}_i^l(\boldsymbol{\theta}', \mathbf{I}), b_i^l(\boldsymbol{\theta}', \mathbf{I})] = [\mathbf{w}_i^l(\boldsymbol{\theta}, \mathbf{I}), b_i^l(\boldsymbol{\theta}, \mathbf{I}) + \delta]$. Putting everything together, we have that

$$c\mathbf{v} = [\mathbf{w}_i^l(\boldsymbol{\theta}, \mathbf{I}), b_i^l(\boldsymbol{\theta}, \mathbf{I})], \quad (22)$$

$$c'\mathbf{v} = [\mathbf{w}_i^l(\boldsymbol{\theta}, \mathbf{I}), b_i^l(\boldsymbol{\theta}, \mathbf{I}) + \delta], \quad (23)$$

which implies $(c' - c)\mathbf{v} = [\mathbf{0}, \delta]$, and in particular $\mathbf{w}_j^k(\boldsymbol{\theta}, \mathbf{J}) = \mathbf{0}$. This contradicts the assumption that $\boldsymbol{\theta} \in \boldsymbol{\Theta}^*$ and completes the proof for the case $(l, i) \neq (k, j)$.

If $(l, i) = (k, j)$, then it must be that $\mathbf{I}_{1:l-1} \neq \mathbf{J}_{1:l-1}$. Wlog, let $(\lambda, \iota) \in [l-1] \times [d_\lambda]$ be such that $I_\lambda[\iota, \iota] = 1$ and $J_\lambda[\iota, \iota] = 0$. Then there exist $\boldsymbol{\theta}, \boldsymbol{\theta}' \in \boldsymbol{\Theta}'$ such that $\boldsymbol{\theta} = (\mathbf{W}_1, \mathbf{b}_1, \ldots, \mathbf{W}_L, \mathbf{b}_L)$ and $\boldsymbol{\theta}' = (\mathbf{W}_1, \mathbf{b}_1, \ldots, \mathbf{W}_\lambda, \mathbf{b}_\lambda + \delta \mathbf{e}_\iota, \ldots, \mathbf{W}_L, \mathbf{b}_L)$, where $\delta \neq 0$. Then there exist $c, c' \in \mathbb{R}$ such that

$$[\mathbf{w}_i^l(\boldsymbol{\theta}, \mathbf{I}), b_i^l(\boldsymbol{\theta}, \mathbf{I})] = c[\mathbf{w}_i^l(\boldsymbol{\theta}, \mathbf{J}), b_i^l(\boldsymbol{\theta}, \mathbf{J})], \tag{24}$$

$$[\mathbf{w}_i^l(\boldsymbol{\theta}', \mathbf{I}), b_i^l(\boldsymbol{\theta}', \mathbf{I})] = c'[\mathbf{w}_i^l(\boldsymbol{\theta}', \mathbf{J}), b_i^l(\boldsymbol{\theta}', \mathbf{J})], \tag{25}$$

where as before, $\mathbf{w}_i^l$ does not depend on the $b_\lambda[\iota]$-component. For $b_i^l$ we now have $b_i^l(\boldsymbol{\theta}', \mathbf{J}) = b_i^l(\boldsymbol{\theta}, \mathbf{J})$ and $b_i^l(\boldsymbol{\theta}', \mathbf{I}) = b_i^l(\boldsymbol{\theta}, \mathbf{I}) + d$, where $d = \delta \mathbf{e}_\iota^\mathsf{T} \mathbf{W}_l \mathbf{I}_{l-1} \cdots \mathbf{W}_{\lambda+1} \mathbf{e}_\iota$ and by membership in $\boldsymbol{\Theta}^*$, $d \neq 0$. From here, we can proceed as in the case $(l, i) \neq (k, j)$, completing the proof for $\boldsymbol{\Theta}^\neg$ being zero-measure.

Finally, we show that $\boldsymbol{\Theta}^\neg$ is closed. Let $\boldsymbol{\theta} \in \boldsymbol{\Theta}^* \setminus \boldsymbol{\Theta}^\neg$, i.e. for all $((l, i, \mathbf{I}), (k, j, \mathbf{J})) \in \mathcal{C}$, the vectors $[\mathbf{w}_i^l(\boldsymbol{\theta}, \mathbf{I}), b_i^l(\boldsymbol{\theta}, \mathbf{I})]$ and $[\mathbf{w}_j^k(\boldsymbol{\theta}, \mathbf{J}), b_j^k(\boldsymbol{\theta}, \mathbf{J})]$ are non-colinear. Since $\mathbf{w}_i^l, b_i^l, \mathbf{w}_j^k, b_j^k$ are continuous functions in $\boldsymbol{\theta}$, there exists a small enough $\epsilon > 0$ such that $[\mathbf{w}_i^l(\boldsymbol{\theta}', \mathbf{I}), b_i^l(\boldsymbol{\theta}', \mathbf{I})]$ and $[\mathbf{w}_j^k(\boldsymbol{\theta}', \mathbf{J}), b_j^k(\boldsymbol{\theta}', \mathbf{J})]$ are non-colinear for all $\boldsymbol{\theta}' \in \mathcal{B}_\epsilon(\boldsymbol{\theta})$ and all $((l, i, \mathbf{I}), (k, j, \mathbf{J})) \in \mathcal{C}$. Hence, $\boldsymbol{\Theta}^* \setminus \boldsymbol{\Theta}^\neg$ is open, and $\boldsymbol{\Theta}^\neg$ is closed. □

**Lemma A.12.** *For all ReLU nets $h : \mathcal{Z} \to \mathbb{R}$ except a closed zero-measure set,*

$$\mathcal{F}_i^l(h) = \left\{ \mathbf{z} \in \mathcal{Z} \mid h_i^{1:l}(\mathbf{z}) = 0 \right\} \tag{26}$$

$$= \left\{ \mathbf{z} \in \mathcal{Z} \mid h_i^{1:l} \text{ is positive and negative in the neighbourhood of } \mathbf{z} \right\} \tag{27}$$

*for all units $(l, i)$.*

*Proof.* We provide a proof for a single unit $(l, i)$; the extension to all units follows from the finite number of units.

Let $\mathcal{G}, \mathcal{H}$, denote the sets defined on the right-hand sides of (26) and (27) respectively. Clearly, $\mathcal{H} \subseteq \mathcal{F}_i^l(h) \subseteq \mathcal{G}$. We will show that $\mathcal{G} \subseteq \mathcal{H}$. Let $\mathcal{Y} \subseteq \mathbb{R}$ denote the set of all local optima of the function $\mathbf{z} \mapsto \mathbf{W}_l[i, :] \cdot \check{h}^{1:l-1}(\mathbf{z})$. Due to piece-wise linearity of the function, and the finite number of pieces, $\mathcal{Y}$ is finite. It follows that for all ReLU networks except a closed zero-measure set, $-b_l[i] \notin \mathcal{Y}$. It is thus guaranteed that $h_i^{1:l}$ never attains a local maximum or minimum at zero. No $\mathbf{z} \in \mathcal{G}$ can therefore be a local maximum or minimum, implying that $h_i^{1:l}$ is both positive and negative in the neighbourhood of $\mathbf{z}$. Hence, $\mathbf{z} \in \mathcal{H}$. □

**Definition A.11** (General ReLU network). *A ReLU network is* general *if it satisfies Lemmas A.10, A.11 and A.12.*

All ReLU networks except a closed zero-measure set are general.

**Lemma A.13.** *If $h$ is a general ReLU network, then $\mathcal{F}(h_i^{1:l}) = \bigcup_{j=1}^{d_{l-1}} \mathcal{F}(\check{h}_j^{1:l-1})$ for all $(l, i)$.*

*Proof.* The inclusion $\mathcal{F}(h_i^{1:l}) \subseteq \bigcup_{j=1}^{d_{l-1}} \mathcal{F}(\check{h}_j^{1:l-1})$ follows from Lemma A.9. For the other inclusion, let $\mathbf{z} \in \mathcal{F}(\check{h}_k^{1:l-1})$ for some $k \in [d_{l-1}]$. Then there exist sequences of points $\mathbf{z}_1(\epsilon), \mathbf{z}_2(\epsilon) \in \mathcal{B}_\epsilon(\mathbf{z}) \setminus \bigcup_{j=1}^{d_{l-1}} \mathcal{F}(\check{h}_j^{1:l-1})$ such that $\mathbf{I}(\mathbf{z}_1(\epsilon)) =: \mathbf{I}$ and $\mathbf{I}(\mathbf{z}_2(\epsilon)) =: \mathbf{J}$ are independent of $\epsilon$, and $\nabla \check{h}_k^{1:l-1}(\mathbf{z}_1(\epsilon)) \neq \nabla \check{h}_k^{1:l-1}(\mathbf{z}_2(\epsilon))$. We consider three cases based on the (non-)triviality of $\mathbf{I}$ and $\mathbf{J}$.

First, suppose both $\mathbf{I}$ and $\mathbf{J}$ are trivial up to $l - 1$. Then by Lemma A.7,

$$\nabla \check{h}_k^{1:l-1}(\mathbf{z}_1(\epsilon)) = I_{l-1}[k, k] \mathbf{w}_k^{l-1}(\mathbf{I}) = \mathbf{0}, \tag{28}$$

and similarly $\nabla \check{h}_k^{1:l-1}(\mathbf{z}_2(\epsilon)) = \mathbf{0}$, which contradicts $\nabla \check{h}_k^{1:l-1}(\mathbf{z}_1(\epsilon) \neq \nabla \check{h}_k^{1:l-1}(\mathbf{z}_2(\epsilon)$. Hence, at least one of $\mathbf{I}, \mathbf{J}$, must be non-trivial up to $l - 1$.

Second, say both $\mathbf{I}$ and $\mathbf{J}$ are non-trivial up to $l - 1$. From $\nabla \check{h}_k^{1:l-1}(\mathbf{z}_1(\epsilon)) \neq \nabla \check{h}_k^{1:l-1}(\mathbf{z}_2(\epsilon))$ it follows that $\mathbf{I}_{1:l-1} \neq \mathbf{J}_{1:l-1}$, we can therefore apply Lemma A.11 to $(l, i, \mathbf{I})$ and $(l, i, \mathbf{J})$. We obtain $\mathbf{w}_i^l(\mathbf{I}) \neq \mathbf{w}_i^l(\mathbf{J})$, implying $\nabla h_i^{1:l}(\mathbf{z}_1(\epsilon)) \neq \nabla h_i^{1:l}(\mathbf{z}_2(\epsilon))$. Thus, $\mathbf{z}$ must be a fold-point of $h_i^{1:l}$.

Finally, say $\mathbf{I}$ is trivial up to $l - 1$ and $\mathbf{J}$ is non-trivial up to $l - 1$. Then $\nabla h_i^{1:l}(\mathbf{z}_1(\epsilon)) = \mathbf{w}_i^l(\mathbf{I}) = \mathbf{0}$, whereas Lemma A.11 applied to $(l, i, \mathbf{J})$ with $c = 0$ yields $\nabla h_i^{1:l}(\mathbf{z}_2(\epsilon)) = \mathbf{w}_i^l(\mathbf{J}) \neq \mathbf{0}$. Hence, $\nabla h_i^{1:l}(\mathbf{z}_1(\epsilon)) \neq \nabla h_i^{1:l}(\mathbf{z}_2(\epsilon))$ and $\mathbf{z}$ must be a fold-point of $h_i^{1:l}$. □

**Definition A.12** (Transparent ReLU network). *A ReLU network $h : \mathcal{Z} \to \mathbb{R}$ is called* transparent up to layer $m$, *if for all $\mathbf{z} \in \mathcal{Z}$ and $l \in [m]$, there exists $i \in [d_l]$ such that $h_i^{1:l}(\mathbf{z}) \geq 0$, or in other words,* $\operatorname{rank}(\mathbf{I}_l(\mathbf{z})) \geq 1$. *If $h$ is transparent up to layer $L-1$, we say it is* transparent.

**Lemma A.14.** *Let $h : \mathcal{Z} \to \mathbb{R}$ be a ReLU network, and let $\lambda \in [L]$. If $h$ is general, then $h^{\lambda:L}|_{\operatorname{int} \mathcal{Z}^{\lambda-1}}$ is general. If $h$ is transparent, then $h^{\lambda:L}|_{\operatorname{int} \mathcal{Z}^{\lambda-1}}$ is transparent.*

*Proof.* We will abbreviate $h^{\lambda:L}|_{\operatorname{int} \mathcal{Z}^{\lambda-1}}$ as $h^{\lambda:L}$. Assume $h$ is general. Then $h^{\lambda:L}$ clearly satisfies Lemma A.10, and for all $(l, i)$, $\mathbf{W}_l[i, :] \neq \mathbf{0}^\intercal$. Next, we prove that $h^{\lambda:L}$ satisfies Lemma A.11. Suppose this was *not* the case; then there exist units $(\lambda - 1 + l, i), (\lambda - 1 + k, j)$, and non-trivial activation indicators $\mathbf{I} = (\mathbf{I}_\lambda, \ldots, \mathbf{I}_{\lambda-1+l})$, $\mathbf{J} = (\mathbf{J}_\lambda, \ldots, \mathbf{J}_{\lambda-1+k})$, with $(l, i, \mathbf{I}) \neq (k, j, \mathbf{J})$, and a scalar $C \in \mathbb{R}$ such that

$$\mathbf{e}_i^\intercal \mathbf{W}_{\lambda-1+l} \mathbf{I}_{\lambda-2+l} \cdots \mathbf{I}_\lambda \mathbf{W}_\lambda = C \cdot \mathbf{e}_j^\intercal \mathbf{W}_{\lambda-1+k} \mathbf{J}_{\lambda-2+k} \cdots \mathbf{J}_\lambda \mathbf{W}_\lambda, \tag{29}$$

and

$$\mathbf{e}_i^\intercal \sum_{l'=\lambda}^{\lambda-1+l} \mathbf{W}_{\lambda-1+l} \mathbf{I}_{\lambda-2+l} \cdots \mathbf{W}_{l'+1} \mathbf{I}_{l'} \mathbf{b}_{l'} = \tag{30}$$

$$C \cdot \mathbf{e}_j^\intercal \sum_{k'=\lambda}^{\lambda-1+k} \mathbf{W}_{\lambda-1+k} \mathbf{J}_{\lambda-2+k} \cdots \mathbf{W}_{k'+1} \mathbf{I}_{k'} \mathbf{b}_{k'}. \tag{31}$$

Then for any non-trivial indicator $(\mathbf{I}_1, \ldots, \mathbf{I}_{\lambda-1}) \triangleq (\mathbf{J}_1, \ldots, \mathbf{J}_{\lambda-1})$, we obtain by post-multiplying (29),

$$\mathbf{e}_i^\intercal \mathbf{W}_{\lambda-1+l} \mathbf{I}_{\lambda-2+l} \cdots \mathbf{I}_1 \mathbf{W}_1 = C \cdot \mathbf{e}_j^\intercal \mathbf{W}_{\lambda-1+k} \mathbf{J}_{\lambda-2+k} \cdots \mathbf{J}_1 \mathbf{W}_1, \tag{32}$$

and for all $\iota \in [\lambda - 1]$,

$$\mathbf{e}_i^\intercal \mathbf{W}_{\lambda-1+l} \mathbf{I}_{\lambda-2+l} \cdots \mathbf{W}_{\iota+1} \mathbf{I}_\iota \mathbf{b}_\iota = C \cdot \mathbf{e}_j^\intercal \mathbf{W}_{\lambda-1+k} \mathbf{J}_{\lambda-2+k} \cdots \mathbf{W}_{\iota+1} \mathbf{J}_\iota \mathbf{b}_\iota. \tag{33}$$

The first equality means that $\mathbf{w}_i^l(\mathbf{I}) = C \cdot \mathbf{w}_j^k(\mathbf{J})$, and the second equality implies $b_i^l(\mathbf{I}) = C \cdot b_j^k(\mathbf{J})$. However, that contradicts the fact that $h$ satisfies Lemma A.11.

The last condition of generality is Lemma A.12. Suppose $h^{\lambda:L}$ does *not* satisfy the lemma. Then there exists a unit $(l, i)$ such that

$$\big\{ \mathbf{z} \in \operatorname{int} \mathcal{Z}^{l-1} \,|\, h_i^{\lambda:l}(\mathbf{z}) = 0 \big\} \nsubseteq$$
$$\big\{ \mathbf{z} \in \operatorname{int} \mathcal{Z}^{l-1} \,|\, h_i^{\lambda:l} \text{ is positive and negative in the neighbourhood of } \mathbf{z} \big\},$$

i.e. there exists $\mathbf{z} \in \operatorname{int} \mathcal{Z}^{l-1}$ such that $h_i^{\lambda:l}(\mathbf{z}) = 0$, and for some $\epsilon > 0$ either $h_i^{\lambda:l}(\mathcal{B}_\epsilon(\mathbf{z})) \subseteq (-\infty, 0]$ or $h_i^{\lambda:l}(\mathcal{B}_\epsilon(\mathbf{z})) \subseteq [0, \infty)$. However, then there exists $\mathbf{z}' \in \mathcal{Z}$ such that $\check{h}^{1:l-1}(\mathbf{z}') = \mathbf{z}$, and for $\mathbf{z}'$ we obtain $h_i^{1:l}(\mathbf{z}') = 0$, and by continuity, there is $\delta > 0$ such that either $h_i^{1:l}(\mathcal{B}_\delta(\mathbf{z}')) \subseteq (-\infty, 0]$ or $h_i^{1:l}(\mathcal{B}_\delta(\mathbf{z}')) \subseteq [0, \infty)$. This contradicts the fact that $h$ satisfies Lemma A.12. We have thus shown that if $h$ is general, then $h^{\lambda:L}|_{\operatorname{int} \mathcal{Z}^{\lambda-1}}$ is general.

Finally, assume $h$ is transparent, i.e. for all $\mathbf{z} \in \mathcal{Z}$ and $l \in [L-1]$, there exists $i \in [d_l]$ such that $h_i^{1:l}(\mathbf{z}) \geq 0$. Then also for all $\mathbf{z} \in \operatorname{int} \mathcal{Z}^{\lambda-1}$ and $l \in \{\lambda, \ldots, L-1\}$, there exists $i \in [d_l]$ such that $h_i^{\lambda:l}(\mathbf{z}) \geq 0$. Hence, $h^{\lambda:L}$ is transparent. $\square$

**Lemma A.15.** *a) For all ReLU networks $h : \mathcal{Z} \to \mathbb{R}$ and all $l \in [L]$, $i \in [d_l]$, we have $\mathcal{F}(h_i^{1:l}) \subseteq \bigcup_{k \in [l-1], j \in [d_k]} \mathcal{F}_j^k(h)$. In particular, $\mathcal{F}(h) \subseteq \bigcup_{k \in [L-1], j \in [d_l]} \mathcal{F}_j^k(h)$.*

*b) For all general ReLU networks $h : \mathcal{Z} \to \mathbb{R}$ transparent up to layer $l - 1$, we have $\mathcal{F}(h_i^{1:l}) = \bigcup_{k \in [l-1], j \in [d_k]} \mathcal{F}_j^k(h)$. In particular, for all general transparent ReLU networks, $\mathcal{F}(h) = \bigcup_{k \in [L-1], j \in [d_k]} \mathcal{F}_j^k(h)$.*

*Proof.* We give a proof of b) only. A proof of a) can be obtained by replacing some equalities by inclusions. We will prove by induction that $\mathcal{F}(h_i^{1:l}) = \bigcup_{k \in [l-1], j \in [d_k]} \mathcal{F}_j^k(h)$ if $h$ is general and transparent up to layer $l - 1$. For $l = 1$, the function $h_i^{1:l}$ is linear, so $\mathcal{F}(h) = \emptyset$ and the claim holds

trivially. Now assume that $\mathcal{F}(h_i^{1:l}) = \bigcup_{k \in [l-1], j \in [d_k]} \mathcal{F}_j^k(h)$ holds; we will prove the same statement for $l+1$. By Lemma A.8 and Lemma A.13, we have

$$\mathcal{F}(\check{h}_i^{1:l}) = \left(\{\mathbf{z} \in \mathcal{Z} \mid h_i^{1:l}(\mathbf{z}) > 0\} \cap \mathcal{F}(h_i^{1:l})\right) \cup \mathcal{F}_i^l(h) \tag{34}$$

$$= \left(\{\mathbf{z} \in \mathcal{Z} \mid h_i^{1:l}(\mathbf{z}) > 0\} \cap \bigcup_{k \in [l-1], j \in [d_k]} \mathcal{F}_j^k(h)\right) \cup \mathcal{F}_i^l(h), \tag{35}$$

$$\mathcal{F}(h_\iota^{1:l+1}) = \bigcup_{i=1}^{d_l} \left(\{\mathbf{z} \in \mathcal{Z} \mid h_i^{1:l}(\mathbf{z}) > 0\} \cap \bigcup_{k \in [l-1], j \in [d_k]} \mathcal{F}_j^k(h)\right) \cup \mathcal{F}_i^l(h) \tag{36}$$

$$= \left(\bigcup_{i=1}^{d_l} \{\mathbf{z} \in \mathcal{Z} \mid h_i^{1:l}(\mathbf{z}) > 0\} \cap \bigcup_{k \in [l-1], j \in [d_k]} \mathcal{F}_j^k(h)\right) \cup \bigcup_{i=1}^{d_l} \mathcal{F}_i^l(h). \tag{37}$$

Since $\bigcup_{i=1}^{d_l} \{\mathbf{z} \in \mathcal{Z} \mid h_i^{1:l}(\mathbf{z}) > 0\} \subseteq \mathcal{Z}$, we obtain

$$\mathcal{F}(h_\iota^{1:l+1}) \subseteq \bigcup_{k \in [l-1], j \in [d_k]} \mathcal{F}_j^k(h) \cup \bigcup_{j \in [d_l]} \mathcal{F}_j^l(h) = \bigcup_{k \in [l], j \in [d_k]} \mathcal{F}_j^k(h). \tag{38}$$

It remains to show the reverse inclusion; we do so by contradiction.

Suppose $\bar{\mathbf{z}} \in \bigcup_{k \in [l], j \in [d_k]} \mathcal{F}_j^k(h) \setminus \mathcal{F}(h_\iota^{1:l+1})$, or equivalently

$$\bar{\mathbf{z}} \in \bigcup_{k \in [l-1], j \in [d_k]} \mathcal{F}_j^k(h) \cap \left(\mathcal{Z} \setminus \bigcup_{i \in [d_l]} \{\mathbf{z} \in \mathcal{Z} \mid h_i^{1:l}(\mathbf{z}) > 0\}\right) \tag{39}$$

$$= \bigcup_{k \in [l-1], j \in [d_k]} \mathcal{F}_j^k(h) \cap \bigcap_{i \in [d_l]} \{\mathbf{z} \in \mathcal{Z} \mid h_i^{1:l}(\mathbf{z}) \leq 0\}. \tag{40}$$

Because $h$ is transparent, there exists $i \in [d_l] : h_i^{1:l}(\bar{\mathbf{z}}) \geq 0$, so for this $i$ we have $h_i^{1:l}(\bar{\mathbf{z}}) = 0$. However, by Lemma A.12, this implies $\bar{\mathbf{z}} \in \mathcal{F}_i^l(h) \subseteq \mathcal{F}(h_\iota^{1:l+1})$. $\qquad\square$

**Lemma A.16.** *Let $h : \mathcal{Z} \to \mathbb{R}$ be a ReLU network. Then $\mathcal{F}_i^{l+1}(h)$ is a union of piece-wise hyperplanes wrt. $\mathcal{P}_{\mathcal{Z}}(\bigcup_{k \in [l], j \in [d_k]} \mathcal{F}_j^k(h))$.*

*Proof.* Since $\mathcal{P}_{\mathcal{Z}}(\mathcal{F}(h_i^{1:l+1}))$ is the partition of the input space into the linear regions of $h_i^{1:l+1}$, and $\mathcal{F}(h_i^{1:l+1}) \subseteq \bigcup_{k \in [l], j \in [d_k]} \mathcal{F}_j^k(h)$ by Lemma A.15, the function $h_i^{1:l+1}$ is also linear on the regions of $\mathcal{P}_{\mathcal{Z}}(\bigcup_{k \in [l], j \in [d_k]} \mathcal{F}_j^k(h))$. For any $P \in \mathcal{P}_{\mathcal{Z}}(\bigcup_{k \in [l], j \in [d_k]} \mathcal{F}_j^k(h))$, denote the slope and bias of $h_i^{1:l+1}$ on $P$ by $\mathbf{w}(P), b(P)$. Then

$$P \cap \mathcal{F}_i^{l+1}(h) = \{\mathbf{z} \in P \mid \mathbf{w}(P)^\mathsf{T}\mathbf{z} + b(P) = 0 \tag{41}$$

$$\text{and } h_i^{1:l+1} \text{ is positive in the neighbourhood of } \mathbf{z}\}. \tag{42}$$

The positivity condition guarantees that $(\mathbf{w}(P), b(P)) \neq (\mathbf{0}, 0)$, so $P \cap \mathcal{F}_i^{l+1}(h)$ is either an empty set or a piece-wise hyperplane. $\qquad\square$

**Corollary A.1.** *Let $h : \mathcal{Z} \to \mathbb{R}$ be a ReLU network. Then the set $\bigcup_{l \in [\kappa], i \in [d_l]} \mathcal{F}_i^l(h)$ is a pwl. surface of order at most $\kappa$. In particular, if $h$ is general and transparent, then $\mathcal{F}(h) = \bigcup_{l \in [L-1], i \in [d_l]} \mathcal{F}_i^l(h)$ is a pwl. surface of order at most $L-1$.*

## A.4 MAIN RESULT

**Lemma A.17.** *For any bounded domain $\mathcal{X}$ and any architecture $(d_1, \ldots, d_{L-1})$ with $d_0 \geq d_1 \geq \cdots \geq d_{L-1} \geq 2$, there exists a nonempty open set of transparent ReLU networks $h : \mathcal{X} \to \mathbb{R}$ such that for $l \in [L-1]$,*

- *$\dim \mathcal{Z}^l = d_l$, and*

- *the set $\mathcal{F}(h^{l:L}|_{\mathrm{int}\,\mathcal{Z}^{l-1}})$ is a pwl. surface whose dependency graph contains $d_l$ directed paths of length $(L-1-l)$ with distinct starting vertices.*

*Proof.* We give an explicit construction; we first state it and then we prove its properties. For $l = 1$, we choose the parameters $(\mathbf{W}_l, \mathbf{b}_l)$ as follows. Let $P_1$ be some nonempty open convex subset of $\mathcal{X}$, and define

$$\mathcal{W}_1^l := \left\{ (\mathbf{w}, b) \in \mathbb{R}^{d_{l-1}} \times \mathbb{R} \;\middle|\; \inf_{\mathbf{z} \in P_l} \mathbf{w}^\mathsf{T}\mathbf{z} + b < 0 < \sup_{\mathbf{z} \in P_l} \mathbf{w}^\mathsf{T}\mathbf{z} + b \right\}. \tag{43}$$

Since $P_l$ has at least two elements, the strict separation theorem implies that $\mathcal{W}_1^l$ is nonempty. It is also open. We can therefore choose $(\mathbf{W}_l[1,:], b_l[1])$ from $\mathcal{W}_1^l$, and define

$$\begin{aligned} \mathcal{W}_2^l := \{ (\mathbf{w}, b) \in \mathcal{W}_1^l \mid \mathbf{w}^\mathsf{T}\mathbf{z} + b > 0 \text{ for all } \mathbf{z} \in \mathcal{Z}^l \\ \text{such that } \mathbf{W}_l[1,:]\mathbf{z} + b_l[1] \le 0 \}. \end{aligned} \tag{44}$$

The set $\mathcal{A} := \left\{ \mathbf{z} \in \mathrm{conv\,cl}\,\mathcal{Z}^l \mid \mathbf{W}_l[1,:]\mathbf{z} + b_l[1] \le 0 \right\}$ is nonempty, convex and compact, and by construction there exists $\mathbf{z} \in P_l \setminus \mathcal{A}$. Again, the strict separation theorem implies $\mathcal{W}_2^l$ is nonempty. It is also open by the boundedness of $\mathcal{A}$. We then choose $(\mathbf{W}_l[2,:], b_l[2])$ from $\mathcal{W}_2^l$, and define

$$\bar{Q}_\iota \triangleq \{ \mathbf{z} \in P_l \mid \mathbf{W}_l[i,:]\mathbf{z} + b_l[i] \ge 0 \text{ for all } i \in [\iota] \}. \tag{45}$$

Note that $\mathrm{int}\,\bar{Q}_2$ is nonempty, open and convex. Also, there exist $\mathbf{z}_1, \mathbf{z}_2 \in \bar{Q}_2$ such that

$$\mathbf{W}_l[i,:]\mathbf{z}_i + b_l[i] = 0, \tag{46}$$
$$\mathbf{W}_l[j,:]\mathbf{z}_i + b_l[j] > 0 \text{ for } j \ne i. \tag{47}$$

For $i \in \{3, \dots, d_l\}$, we then choose $(\mathbf{W}_l[i,:], b_l[i])$ as follows. Let $\mathcal{A} := \mathrm{conv}\{\mathbf{z}_1, \dots, \mathbf{z}_{i-1}\}$ and let $\mathbf{z} \in \mathrm{int}\,\bar{Q}_{i-1} \setminus \mathcal{A}$. Such $\mathbf{z}$ exists because $\dim \bar{Q}_{i-1} = \dim P_l = d_{l-1}$, and $\dim \mathcal{A} \le i - 2 \le d_l - 2 \le d_{l-1} - 2$. By strict separation and boundedness, there exists a nonempty open set of hyperplanes that strictly separate $\mathcal{A}$ and $\mathbf{z}$. Let $(\mathbf{W}_l[i,:], b_l[i])$ be any of them, oriented such that $\mathbf{W}_l[i,:]\mathbf{z}_j + b_l[i] > 0$ for $j \in [i-1]$. Then denote by $\mathbf{z}_i \in \mathrm{int}\,\bar{Q}_{i-1}$ any point that satisfies $\mathbf{W}_l[i,:]\mathbf{z}_i + b_l[i] = 0$; it exists by convexity. Then $\mathrm{int}\,\bar{Q}_i$ is nonempty and convex, and the construction for $\{i+1, \dots, d_l\}$ goes through.

For layers $l \in \{2, \dots, L-1\}$, we use a similar construction as for $l = 1$. Denote

$$P_l := (\mathbf{W}_{l-1}P_{l-1} + \mathbf{b}_{l-1}) \cap \{\mathbf{z} > \mathbf{0}\}, \tag{48}$$
$$\bar{P}_l := (\mathbf{W}_{l-1}P_{l-1} + \mathbf{b}_{l-1}) \cap \{\mathbf{z} \ge \mathbf{0}\}. \tag{49}$$

$P_l$ is nonempty (because $\mathrm{int}\,\bar{Q}_{d_{l-1}}$ is nonempty), open (because $\mathbf{W}_{l-1}$ is wide) and convex. We assume $\mathbf{W}_{l-1}$ is full-rank; this holds for all choices of $\mathbf{W}_{l-1}$ except a closed zero-measure set. By the construction of $\bar{Q}_{d_{l-1}}$, there exist $\mathbf{z}_1, \dots, \mathbf{z}_{d_{l-1}} \in \bar{Q}_{d_{l-1}} \subseteq P_{l-1}$ such that or $i \in [d_{l-1}]$,

$$\mathbf{W}_{l-1}[i,:]\mathbf{z}_i + b_{l-1}[i] = 0, \tag{50}$$
$$\mathbf{W}_{l-1}[j,:]\mathbf{z}_i + b_{l-1}[j] > 0 \text{ for } j \ne i. \tag{51}$$

Denote their images $\mathbf{z}_i' := \mathbf{W}_{l-1}\mathbf{z}_i + \mathbf{b}_{l-1}$. Then by openness, there exists a ball around each $\mathbf{z}_i'$ such that

$$\mathcal{B}_\epsilon(\mathbf{z}_i') \subseteq (\mathbf{W}_{l-1}P_{l-1} + \mathbf{b}_{l-1}) \cap \{z'[j] > 0 \text{ for } j \ne i\}, \tag{52}$$

implying that each $\mathbf{z}_i'$ has a $d_{l-1}$-dimensional, relatively open neighbourhood $\mathcal{B}_\epsilon(\mathbf{z}_i') \cap \{z'[i] = 0\} \subseteq \bar{P}_l$ whose elements satisfy $z'[i] = 0$ and $z'[j] > 0$ for $j \ne i$. It follows that the set

$$\begin{aligned} \mathcal{W}_1^l := \Big\{ (\mathbf{w}, b) \in \mathbb{R}^{d_{l-1}} \times \mathbb{R} \;\Big|\; \inf_{\mathbf{z} \in P_l} \mathbf{w}^\mathsf{T}\mathbf{z} + b < 0 < \sup_{\mathbf{z} \in P_l} \mathbf{w}^\mathsf{T}\mathbf{z} + b, \text{ and} \\ \forall i \in [d_{l-1}]\, \exists \mathbf{z} \in \bar{P}_l : z[i] = 0, z[j] > 0 \text{ for } j \ne i, \text{ and } \mathbf{w}^\mathsf{T}\mathbf{z} + b = 0 \Big\} \end{aligned} \tag{53}$$

contains a nonempty open subset. We can therefore choose $(\mathbf{W}_l[1,:], b_l[1])$ from $\mathcal{W}_1^l$. For the choice of $(\mathbf{W}_l[i,:], b_l[i])$ for $i \in \{2, \dots, d_l\}$, we use the same procedure as in the first layer. Finally, choose $(\mathbf{W}_L, b_L)$ arbitrarily.

We will show that this construction satisfies the lemma. The networks are transparent because of how we define $\mathcal{W}_2^l$: for all $\mathbf{x} \in X$ and $l \in [L-1]$, either $h_1^{1:l}(\mathbf{x}) > 0$ or $h_2^{1:l}(\mathbf{x}) > 0$. Also, $\dim \mathcal{Z}^l = d_l$ because $\mathcal{Z}^l$ contains $\operatorname{int} \bar{Q}_{d_l}$, which is nonempty and open.

Now let $l \in [L]$. We can think of the function $h^{l:L}|_{\operatorname{int} \mathcal{Z}^{l-1}}$ as an $(L-l+1)$-layer ReLU network parameterised by $(\mathbf{W}_l, \mathbf{b}_l, \ldots, \mathbf{W}_L, b_L)$. Because $h$ is transparent, also $h^{l:L}|_{\operatorname{int} \mathcal{Z}^{l-1}}$ is transparent. Corollary A.1 then implies that $\mathcal{S} := \mathcal{F}(h^{l:L}|_{\operatorname{int} \mathcal{Z}^{l-1}}) = \bigcup_{k \in [L-l], j \in [d_k]} \mathcal{F}_j^k(h^{l:L}|_{\operatorname{int} \mathcal{Z}^{l-1}})$ is a pwl. surface. Let $\mathcal{S} = \bigcup_{k \in [L-l], j \in [n_\lambda]} \mathcal{H}_j^k$ be its canonical representation, and let $G$ denote its dependency graph.

To find the required paths in $G$, we first identify some important vertices. For $\lambda \in [L-l]$, denote

$$\mathcal{Z}_\lambda^+ := \{\mathbf{z} \in P_l \mid h^{l:l+k-1}(\mathbf{z}) > \mathbf{0} \text{ for } k \in [\lambda-1]\}. \tag{54}$$

This set is nonempty and open because $P_{l+\lambda}$ is nonempty and open. Next, for any unit $(\lambda, \iota)$,

$$\mathcal{F}_\iota^\lambda(h^{l:L}|_{\operatorname{int} \mathcal{Z}^{l-1}}) \cap \mathcal{Z}_\lambda^+ = \{\mathbf{z} \in P_l \mid h_\iota^{l:l+\lambda-1}(\mathbf{z}) = 0,$$
$$h^{l:l+k-1}(\mathbf{z}) > \mathbf{0} \text{ for } k \in [\lambda-1]\}. \tag{55}$$

By the definition of $\mathcal{W}_1^{l+\lambda-1}$ and the fact that $h^{l:l+\lambda-1}(\mathcal{Z}_\lambda^+) = P_{l+\lambda}$, the set $\mathcal{A} := \mathcal{F}_\iota^\lambda(h^{l:L}|_{\operatorname{int} \mathcal{Z}^{l-1}}) \cap \mathcal{Z}_\lambda^+$ is nonempty. Also, by the definition of $\{P_{l'}\}_{l'}$, $h^{l:l+\lambda-1}$ is in fact linear on $\mathcal{Z}_\lambda^+$, so $\mathcal{A}$ is a hyperplane on $\mathcal{Z}_\lambda^+$. Therefore there exists a piece-wise hyperplane $\mathcal{H}_{j(\lambda,\iota)}^{k(\lambda,\iota)}$ from the canonical representation of $\mathcal{S}$ that contains $\mathcal{A}$; by Lemma A.11, all $\{\mathcal{H}_{j(\lambda,\iota)}^{k(\lambda,\iota)}\}_{\lambda,\iota}$ are distinct from each other.

We now show that $G$ contains the edge $\mathcal{H}_{j(\lambda,i)}^{k(\lambda,i)} \to \mathcal{H}_{j(\lambda+1,\iota)}^{k(\lambda+1,\iota)}$ for $\lambda \in [L-l-1]$ and all $(i, \iota)$. By the definition of $\mathcal{W}^{l+\lambda}$, there exists $\bar{z} \in \bar{P}_{l+\lambda}$ such that $\bar{z}[i] = 0$, $\bar{z}[j] > 0$ for $j \neq i$, and $\mathbf{W}_{l+\lambda}[\iota, :]\bar{z} + b_{l+\lambda}[\iota] = 0$. There also exists a ball $\mathcal{B}_\epsilon(\bar{z}) \subseteq (\mathbf{W}_{l+\lambda-1}P_{l+\lambda-1} + \mathbf{b}_{l+\lambda-1}) \cap \{z[j] > 0 \text{ for } j \neq i\}$. Then because of how $\{P_{l'}\}_{l'}$ are defined, there exists $\bar{z}' \in P_l$ such that $h^{l:l+\lambda-1}(\bar{z}') = \bar{z}$, so it satisfies

$$h_i^{l:l+\lambda-1}(\bar{z}') = 0, \tag{56}$$
$$h_j^{l:l+k-1}(\bar{z}') > 0, \text{ for } k \in [\lambda], (k, j) \neq (\lambda, i), \tag{57}$$
$$h_\iota^{l:l+\lambda}(\bar{z}') = 0. \tag{58}$$

It follows that $\bar{z}' \in \mathcal{F}_i^\lambda(h^{l:L}|_{\operatorname{int} \mathcal{Z}^{l-1}}) \cap \mathcal{Z}_\lambda^+ \subseteq \mathcal{H}_{j(\lambda,i)}^{k(\lambda,i)}$. At the same time, the preimage $(h^{l:l+\lambda-1})^{-1}(\mathcal{B}_\epsilon(\bar{z}))$ is open by continuity, and contains $\bar{z}'$. So there exists a ball $\mathcal{B}_\epsilon(\bar{z}') \subseteq P_l$ such that all $\mathbf{z}' \in \mathcal{B}_\epsilon(\bar{z}')$ satisfy

$$h_j^{l:l+k-1}(\mathbf{z}') > 0, \text{ for } k \in [\lambda], (k, j) \neq (\lambda, i). \tag{59}$$

On this ball, $h_i^{l:l+\lambda-1}$ is linear, so the set $\mathcal{A} := \mathcal{B}_\epsilon(\bar{z}') \cap \{h_i^{l:l+\lambda-1}(\mathbf{z}') > 0\}$ is an open half-ball. On $\mathcal{A}$, $h_\iota^{l:l+\lambda}$ is linear as well, and the set $\{\mathbf{z}' \mid h_\iota^{l:l+\lambda}(\mathbf{z}') = 0\}$ intersects the center of the half-ball, $\bar{z}'$. Therefore there exists a sequence of points $\{\mathbf{z}_n'\} \subseteq P_l$ such that $\mathbf{z}_n' \to \bar{z}'$ and

$$h^{l:l+k-1}(\mathbf{z}') > \mathbf{0}, \text{ for } k \in [\lambda], \tag{60}$$
$$h_\iota^{l:l+\lambda}(\mathbf{z}') = 0. \tag{61}$$

We obtain that $\bar{z}' \in \operatorname{cl}(\mathcal{F}_\iota^{\lambda+1}(h^{l:L}|_{\operatorname{int} \mathcal{Z}^{l-1}}) \cap \mathcal{Z}_{\lambda+1}^+) \subseteq \operatorname{cl} \mathcal{H}_{j(\lambda+1,\iota)}^{k(\lambda+1,\iota)}$, which implies

$$\operatorname{cl} \mathcal{H}_{j(\lambda,i)}^{k(\lambda,i)} \cap \operatorname{cl} \mathcal{H}_{j(\lambda+1,\iota)}^{k(\lambda+1,\iota)} \neq \emptyset. \tag{62}$$

It remains to show that $k(\lambda, i) < k(\lambda+1, \iota)$. Consider again the ball $\mathcal{B}_\epsilon(\bar{z}')$. By Lemma A.11, $h_\iota^{l:l+\lambda}$ is a different linear function on $\mathcal{B}_\epsilon(\bar{z}') \cap \{h_i^{l:l+\lambda-1}(\mathbf{z}') > 0\}$ and on $\mathcal{B}_\epsilon(\bar{z}') \cap \{h_i^{l:l+\lambda-1}(\mathbf{z}') < 0\}$. Hence, $\mathcal{H}_{j(\lambda+1,\iota)}^{k(\lambda+1,\iota)}$ is *not* a piece-wise hyperplane wrt. any partition that does not include $\mathcal{H}_{j(\lambda,i)}^{k(\lambda,i)}$. We obtain that $k(\lambda, i) < k(\lambda+1, \iota)$, proving that $G$ contains the edge $\mathcal{H}_{j(\lambda,i)}^{k(\lambda,i)} \to \mathcal{H}_{j(\lambda+1,\iota)}^{k(\lambda+1,\iota)}$.

Finally, observe that the $d_l$ paths $\mathcal{H}_{j(1,i)}^{k(1,i)} \to \mathcal{H}_{j(2,1)}^{k(2,1)} \to \cdots \to \mathcal{H}_{j(L-l,1)}^{k(L-l,1)}$ have length $(L-l-1)$, and distinct starting vertices. This proves the theorem. □

**Lemma A.18.** *For all general ReLU networks $h : \mathcal{Z} \to \mathbb{R}$, the following holds. Denote $\mathcal{S} = \bigcup_{l \in [\lambda], i \in [d_l]} \mathcal{F}_i^l(h)$ and let $\mathcal{S} = \bigcup_{k \in [\kappa], j \in [n_k]} \mathcal{H}_j^k$ be the canonical representation of $\mathcal{S}$. Then $\mathcal{H}_j^k \subseteq \mathcal{F}_i^l(h)$ for some $(l, i)$ with $l \geq k$. Moreover, if the dependency graph of $\mathcal{S}$ contains a directed path of length $m$ starting at $\mathcal{H}_j^k$, then $l \leq \lambda - m$.*

*Proof.* Because the representation is canonical, we have

$$\mathcal{H}_j^k \not\subseteq \square_{k-1} \mathcal{S} \supseteq \bigcup_{l \in [k-1], i \in [d_l]} \mathcal{F}_i^l(h), \tag{63}$$

which implies $\mathcal{H}_j^k \subseteq \bigcup_{l \geq k, i} \mathcal{F}_i^l(h)$. By piece-wise linearity, we can write

$$\bigcup_{l \geq k, i} \mathcal{F}_i^l(h) = \bigcup_{l \geq k, i, P} \mathcal{F}_i^l(h) \cap P = \bigcup_{l \geq k, i, P} \{\mathbf{z} \in P \mid \mathbf{w}_i^l(P) \cdot \mathbf{z} + b_i^l(P) = 0\}, \tag{64}$$

where $P$ runs over the linear regions of $h_i^{1:l}$. Moreover, by the definition of $\mathcal{F}_i^l(h)$, all $\mathbf{w}_i^l(P) \neq \mathbf{0}$. Combined with Lemma A.11, we obtain that each nonempty set on the right-hand side of (64) is a different hyperplane on an open set. Therefore there exists one for which $\mathcal{H}_j^k \subseteq \mathcal{F}_i^l(h) \cap P \subseteq \mathcal{F}_i^l(h)$.

Now assume that the dependency graph of $\mathcal{S}$ contains a directed path of length $m$ starting at $\mathcal{H}_j^k =: \mathcal{H}_{j_0}^{k_0}$; denote the path $\mathcal{H}_{j_0}^{k_0} \to \mathcal{H}_{j_1}^{k_1} \to \cdots \to \mathcal{H}_{j_m}^{k_m}$. By the first part of the lemma, we know that $\mathcal{H}_{j_\iota}^{k_\iota} \subseteq \mathcal{F}_{i_\iota}^{l_\iota}(h)$ for some $l_\iota$. Let $\iota \in [m]$; we will show that $l_{\iota-1} < l_\iota$.

Because of the edge $\mathcal{H}_{j_{\iota-1}}^{k_{\iota-1}} \to \mathcal{H}_{j_\iota}^{k_\iota}$, we know that $\mathcal{H}_{j_{\iota-1}}^{k_{\iota-1}}$ and $\text{cl}\,\mathcal{H}_{j_\iota}^{k_\iota}$ intersect, and that $\mathcal{H}_{j_\iota}^{k_\iota}$ is a piece-wise hyperplane wrt. some partition $\mathcal{P}$ for which $\mathcal{H}_{j_{\iota-1}}^{k_{\iota-1}}$ is a boundary. Let $\hat{\mathbf{z}}$ be any point of intersection. By openness, there exists a ball $\mathcal{B}_\epsilon(\hat{\mathbf{z}})$ such that $\mathcal{H}_{j_{\iota-1}}^{k_{\iota-1}}$ is a hyperplane on $\mathcal{B}_\epsilon(\hat{\mathbf{z}})$, and $\mathcal{H}_{j_\iota}^{k_\iota}$ is a hyperplane on one half-ball defined by $\mathcal{B}_\epsilon(\hat{\mathbf{z}})$ and $\mathcal{H}_{j_{\iota-1}}^{k_{\iota-1}}$. If it was the case that $l_{\iota-1} \geq l_\iota$, then $\mathcal{F}_{i_\iota}^{l_\iota}(h)$ would be a hyperplane on $\mathcal{B}_\epsilon(\hat{\mathbf{z}})$, i.e. there would have to exist some piece-wise hyperplane on the opposite half-ball as $\mathcal{H}_{j_{\iota-1}}^{k_{\iota-1}}$, but included in the same hyperplane. However, by Lemma A.11, no two piece-wise hyperplanes in $\mathcal{S}$ are included in a single hyperplane, so we get a contradiction.

Hence, we obtain $l_0 < l_1 < \cdots < l_m \leq \lambda$, which yields $l_0 \leq \lambda - m$. $\qquad\square$

**Lemma A.19.** *Let $(\mathbf{w}, b), (\mathbf{c}, a) \in \mathbb{R}^d \times \mathbb{R}$ and let $F \subseteq \mathbb{R}^d$ with $\dim F = d - 1$. If $\mathbf{w}^\mathsf{T}\mathbf{z} + b = 0$ and $\mathbf{c}^\mathsf{T}\mathbf{z} + a = 0$ for all $\mathbf{z} \in F$, then either $(\mathbf{w}, b) = (\mathbf{0}, 0)$, $(\mathbf{c}, a) = (\mathbf{0}, 0)$, or there exists $\beta \in \mathbb{R} : (\mathbf{c}, a) = \beta(\mathbf{w}, b)$.*

*Proof.* Since $\dim F = d - 1$, there exist $d$ affinely independent vectors $\mathbf{f}_0, \ldots, \mathbf{f}_{d-1}$ in $F$. Hence there are $d - 1$ linearly independent vectors $\mathbf{v}_1 := \mathbf{f}_1 - \mathbf{f}_0, \ldots, \mathbf{v}_{d-1} := \mathbf{f}_{d-1} - \mathbf{f}_0$, such that $\mathbf{w}^\mathsf{T}\mathbf{v}_i = \mathbf{c}^\mathsf{T}\mathbf{v}_i = 0$. In other words, both $\mathbf{w}$ and $\mathbf{c}$ lie in the orthogonal complement of the span of $\mathbf{v}_1, \ldots, \mathbf{v}_{d-1}$. If $\mathbf{w} = \mathbf{0}$, then necessarily $b = 0$, and similarly for $(\mathbf{c}, a)$. If $\mathbf{w} \neq \mathbf{0} \neq \mathbf{c}$, then because the orthogonal complement is one-dimensional, there exists $\beta \in \mathbb{R}$ such that $\mathbf{c} = \beta\mathbf{w}$. Then $\mathbf{c}^\mathsf{T}\mathbf{z} + a - \beta(\mathbf{w}^\mathsf{T}\mathbf{z} + b) = a - \beta b = 0$ and the lemma follows. $\qquad\square$

**Theorem A.1.** *Consider a bounded domain $\mathcal{X}$ and any architecture $(d_1, \ldots, d_{L-1})$ with $d_0 \geq d_1 \geq \cdots \geq d_{L-1} \geq 2$. Let $h_{\boldsymbol{\theta}} : \mathcal{X} \to \mathbb{R}$ be a general ReLU network satisfying Lemma A.17, and let $h_{\boldsymbol{\eta}} : \mathcal{X} \to \mathbb{R}$ be any general ReLU network such that $h_{\boldsymbol{\theta}}(\mathbf{x}) = h_{\boldsymbol{\eta}}(\mathbf{x})$ for all $\mathbf{x} \in \mathcal{X}$. Denote $\boldsymbol{\eta} \triangleq (\mathbf{W}_1', \mathbf{b}_1', \ldots, \mathbf{W}_L', b_L')$. Then there exist permutation matrices $\mathbf{P}_1, \ldots \mathbf{P}_{L-1}$, and positive-entry diagonal matrices $\mathbf{M}_1, \ldots, \mathbf{M}_{L-1}$, such that*

$$\begin{aligned} \mathbf{W}_1 &= \mathbf{M}_1 \mathbf{P}_1 \mathbf{W}_1', & \mathbf{b}_1 &= \mathbf{M}_1 \mathbf{P}_1 \mathbf{b}_1', \\ \mathbf{W}_l &= \mathbf{M}_l \mathbf{P}_l \mathbf{W}_l' \mathbf{P}_{l-1}^{-1} \mathbf{M}_{l-1}^{-1}, & \mathbf{b}_l &= \mathbf{M}_l \mathbf{P}_l \mathbf{b}_l', & l \in \{2, \ldots, L-1\}, \\ \mathbf{W}_L &= \mathbf{W}_L' \mathbf{P}_{L-1}^{-1} \mathbf{M}_{L-1}^{-1}, & b_L &= b_L'. \end{aligned} \tag{65}$$

*Proof.* First, notice that $h_{\boldsymbol{\eta}}$ is transparent. To see this, observe that $h_{\boldsymbol{\theta}}$ is transparent, i.e. $\mathrm{rank}(\mathbf{I}_l^{\boldsymbol{\theta}}(\mathbf{x})) \geq 1$ for all $l \in [L-1]$ and $\mathbf{x} \in \mathcal{X}$. By Lemma A.10, $\nabla_{\mathbf{x}} h_{\boldsymbol{\eta}}(\mathbf{x}) = \nabla_{\mathbf{x}} h_{\boldsymbol{\theta}}(\mathbf{x}) \neq \mathbf{0}^{\mathsf{T}}$, implying that $h_{\boldsymbol{\eta}}$ is transparent.

We proceed by induction. Let $l = 1$. Then we have

$$h_{\boldsymbol{\theta}}^{l:L}\big|_{\mathrm{int}\,\mathcal{Z}_{\boldsymbol{\theta}}^{l-1}} \equiv h_{\boldsymbol{\theta}} \equiv h_{\boldsymbol{\eta}} \equiv h_{\boldsymbol{\eta}}^{l:L}\big|_{\mathrm{int}\,\mathcal{Z}_{\boldsymbol{\theta}}^{l-1}} \tag{66}$$

which implies $\mathcal{F}(h_{\boldsymbol{\theta}}^{l:L}\big|_{\mathrm{int}\,\mathcal{Z}_{\boldsymbol{\theta}}^{l-1}}) = \mathcal{F}(h_{\boldsymbol{\eta}}^{l:L}\big|_{\mathrm{int}\,\mathcal{Z}_{\boldsymbol{\theta}}^{l-1}})$. (For notational convenience, we will omit the domain restriction for now.) Because both networks are general and transparent, Corollary A.1 implies that the set

$$\bigcup_{k \in [L-l], j \in [d_k]} \mathcal{F}_j^k(h_{\boldsymbol{\theta}}^{l:L}) = \mathcal{F}(h_{\boldsymbol{\theta}}^{l:L}) = \mathcal{F}(h_{\boldsymbol{\eta}}^{l:L}) = \bigcup_{k \in [L-l], j \in [d_k]} \mathcal{F}_j^k(h_{\boldsymbol{\eta}}^{l:L}) \tag{67}$$

is a pwl. surface of order at most $L-l$. By Lemma A.17, its graph contains $d_l$ directed paths of length $(L-1-l)$ with distinct starting vertices. Denote these vertices $\mathcal{H}_1, \ldots, \mathcal{H}_{d_l}$. By Lemma A.18, $\mathcal{H}_i \subseteq \mathcal{F}_\iota^\lambda(h_{\boldsymbol{\theta}}^{l:L})$ for some $(\lambda, \iota)$ with $\lambda \leq (L-l) - (L-1-l) = 1$. We thus obtain $\bigcup_{i \in [d_l]} \mathcal{H}_i \subseteq \bigcup_{i \in [d_l]} \mathcal{F}_i^1(h_{\boldsymbol{\theta}}^{l:L})$, where on the left-hand side we have a union of $d_l$ hyperplanes, and on the right-hand side we have a union of at most $d_l$ hyperplanes. It follows that $\bigcup_{i \in [d_l]} \mathcal{H}_i = \bigcup_{i \in [d_l]} \mathcal{F}_i^1(h_{\boldsymbol{\theta}}^{l:L})$, and by applying the same argument to $h_{\boldsymbol{\eta}}$, we get $\bigcup_{i \in [d_l]} \mathcal{F}_i^1(h_{\boldsymbol{\theta}}^{l:L}) = \bigcup_{i \in [d_l]} \mathcal{F}_i^1(h_{\boldsymbol{\eta}}^{l:L})$. Therefore there must exist a permutation $\pi : [d_l] \to [d_l]$ such that $\mathcal{F}_i^1(h_{\boldsymbol{\theta}}^{l:L}) = \mathcal{F}_{\pi(i)}^1(h_{\boldsymbol{\eta}}^{l:L})$ for all $i$. Then by Lemma A.19, there exist scalars $m_1, \ldots m_{d_l}$, such that

$$(\mathbf{W}_l[i,:], b_l[i]) = m_i(\mathbf{W}_l'[\pi(i),:], b_l'[\pi(i)]). \tag{68}$$

We know that $m_i \neq 0$ because the folds $\mathcal{F}_i^1(h_{\boldsymbol{\theta}}^{l:L}), \mathcal{F}_i^1(h_{\boldsymbol{\eta}}^{l:L})$, are nonempty; otherwise $\bigcup_{i \in [d_l]} \mathcal{H}_i$ could not be a union of $d_l$ hyperplanes. We have thus shown that there exists a permutation matrix $\mathbf{P}_l \in \mathbb{R}^{d_l \times d_l}$ and a nonzero-entry diagonal matrix $\mathbf{M}_l \in \mathbb{R}^{d_l \times d_l}$ such that $\mathbf{W}_l = \mathbf{M}_l \mathbf{P}_l \mathbf{W}_l'$ and $\mathbf{b}_l = \mathbf{M}_l \mathbf{P}_l \mathbf{b}_l'$.

Next, we show that the diagonal entries of $\mathbf{M}_l$ are positive. Let $\mathbf{z}^-, \mathbf{z}^+ \in \mathrm{int}\,\mathcal{Z}_{\boldsymbol{\theta}}^{l-1}$ be such that $\mathbf{I}^{\boldsymbol{\theta}}(\mathbf{z}^-)$ and $\mathbf{I}^{\boldsymbol{\theta}}(\mathbf{z}^+)$ differ only in $I_l^{\boldsymbol{\theta}}[i,i]$. Wlog, let $I_l^{\boldsymbol{\theta}}[i,i](\mathbf{z}^-) = 0$ and $I_l^{\boldsymbol{\theta}}[i,i](\mathbf{z}^+) = 1$, and denote the row span of $\mathbf{I}_l^{\boldsymbol{\theta}}(\mathbf{z}^-)\mathbf{W}_l$ by $\mathcal{W}$. Then

$$\nabla_{\mathbf{z}} h_{\boldsymbol{\theta}}^{l:L}(\mathbf{z}^-) = \mathbf{W}_L \mathbf{I}_{L-1}^{\boldsymbol{\theta}}(\mathbf{z}^-) \cdots \mathbf{W}_{l+1} \mathbf{I}_l^{\boldsymbol{\theta}}(\mathbf{z}^-) \mathbf{W}_l \in \mathcal{W},$$
$$\nabla_{\mathbf{z}} h_{\boldsymbol{\theta}}^{l:L}(\mathbf{z}^+) = \mathbf{W}_L \mathbf{I}_{L-1}^{\boldsymbol{\theta}}(\mathbf{z}^+) \cdots \mathbf{W}_{l+1} \mathbf{I}_l^{\boldsymbol{\theta}}(\mathbf{z}^+) \mathbf{W}_l \in \mathrm{span}(\mathcal{W} \cup \mathbf{W}_l[i,:]).$$

Since $h_{\boldsymbol{\theta}}$ is general and $d_l \leq d_{l-1}$, the matrix $\mathbf{W}_l$ has full row rank. This means that the rows of $\mathbf{W}_l$ form a basis, in which the representation of $\nabla_{\mathbf{z}} h_{\boldsymbol{\theta}}^{l:L}(\mathbf{z}^-)$ has one more zero coefficient compared to $\nabla_{\mathbf{z}} h_{\boldsymbol{\theta}}^{l:L}(\mathbf{z}^+)$. In other words, for two points $\mathbf{z}^-, \mathbf{z}^+ \in \mathrm{int}\,\mathcal{Z}_{\boldsymbol{\theta}}^{l-1}$ whose indicators differ only in $I_l^{\boldsymbol{\theta}}[i,i]$, the point for which $I_l^{\boldsymbol{\theta}}[i,i](\mathbf{z}) = 0$ is also the one for which $\nabla_{\mathbf{z}} h_{\boldsymbol{\theta}}^{l:L}(\mathbf{z})$ has more zero coefficients when expressed in the row basis of $\mathbf{W}_l$. Now, observe that if $\mathbf{I}^{\boldsymbol{\theta}}(\mathbf{z}^-)$ and $\mathbf{I}^{\boldsymbol{\theta}}(\mathbf{z}^+)$ differ only in $I_l^{\boldsymbol{\theta}}[i,i]$, then $\mathbf{I}^{\boldsymbol{\eta}}(\mathbf{z}^-)$ and $\mathbf{I}^{\boldsymbol{\eta}}(\mathbf{z}^+)$ differ only in $I_l^{\boldsymbol{\eta}}[\pi(i), \pi(i)]$. Because $\mathbf{W}_l = \mathbf{M}_l \mathbf{P}_l \mathbf{W}_l'$, the number of zero coefficients of $\nabla_{\mathbf{z}} h_{\boldsymbol{\eta}}^{l:L}(\mathbf{z}^-) = \nabla_{\mathbf{z}} h_{\boldsymbol{\theta}}^{l:L}(\mathbf{z}^-)$ in the row basis of $\mathbf{W}_l'$ is the same as the number of zero coefficients of $\nabla_{\mathbf{z}} h_{\boldsymbol{\theta}}^{l:L}(\mathbf{z}^-)$ in the row basis of $\mathbf{W}_l$. It follows that $I_l^{\boldsymbol{\eta}}[\pi(i), \pi(i)](\mathbf{z}^-) = 0$ and $I_l^{\boldsymbol{\eta}}[\pi(i), \pi(i)](\mathbf{z}^+) = 1$. Hence, $m_i$ is positive.

For the inductive step, let $l \in \{2, \ldots, L-1\}$, and assume that there exist permutation matrices $\mathbf{P}_1, \ldots, \mathbf{P}_{l-1}$, and positive-entry diagonal matrices $\mathbf{M}_1, \ldots, \mathbf{M}_{l-1}$, such that (65) holds up to layer $l-1$. Then $h_{\boldsymbol{\theta}}^{1:l-1} \equiv \mathbf{M}_{l-1} \mathbf{P}_{l-1} h_{\boldsymbol{\eta}}^{1:l-1}$. Since $h_{\boldsymbol{\theta}}^{1:L} \equiv h_{\boldsymbol{\eta}}^{1:L}$, it follows that

$$h_{\boldsymbol{\theta}}^{l:L}\big|_{\mathrm{int}\,\mathcal{Z}_{\boldsymbol{\theta}}^{l-1}} \equiv \left(h_{\boldsymbol{\eta}}^{l:L} \circ \mathbf{P}_{l-1}^{-1} \mathbf{M}_{l-1}^{-1}\right)\big|_{\mathrm{int}\,\mathcal{Z}_{\boldsymbol{\theta}}^{l-1}} \equiv h_{\tilde{\boldsymbol{\eta}}}^{l:L}\big|_{\mathrm{int}\,\mathcal{Z}_{\boldsymbol{\theta}}^{l-1}}, \tag{69}$$

where $\tilde{\boldsymbol{\eta}} := (\mathbf{W}_l' \mathbf{P}_{l-1}^{-1} \mathbf{M}_{l-1}^{-1}, \mathbf{b}_l', \mathbf{W}_{l+1}', \mathbf{b}_{l+1}', \ldots, \mathbf{W}_L', \mathbf{b}_L')$. We can therefore apply the same argument to $h_{\boldsymbol{\theta}}^{l:L}\big|_{\mathrm{int}\,\mathcal{Z}_{\boldsymbol{\theta}}^{l-1}}$ and $h_{\tilde{\boldsymbol{\eta}}}^{l:L}\big|_{\mathrm{int}\,\mathcal{Z}_{\boldsymbol{\theta}}^{l-1}}$ as we presented above for the case $l = 1$. We obtain that there exists a permutation matrix $\mathbf{P}_l \in \mathbb{R}^{d_l \times d_l}$ and a positive-entry diagonal matrix $\mathbf{M}_l \in \mathbb{R}^{d_l \times d_l}$ such that

$$\mathbf{W}_l = \mathbf{M}_l \mathbf{P}_l \mathbf{W}_l' \mathbf{P}_{l-1}^{-1} \mathbf{M}_{l-1}^{-1}, \qquad \mathbf{b}_l = \mathbf{M}_l \mathbf{P}_l \mathbf{b}_l'. \tag{70}$$

Finally, consider the last layer. We know that $h_{\boldsymbol{\theta}}^{1:L-1} \equiv \mathbf{M}_{L-1}\mathbf{P}_{L-1}h_{\boldsymbol{\eta}}^{1:L-1}$, which implies $h_{\boldsymbol{\theta}}^{L} \equiv h_{\boldsymbol{\eta}}^{L} \circ \mathbf{P}_{L-1}^{-1}\mathbf{M}_{L-1}^{-1}$, i.e. $h_{\boldsymbol{\theta}}^{L}$ and $h_{\boldsymbol{\eta}}^{L} \circ \mathbf{P}_{L-1}^{-1}\mathbf{M}_{L-1}^{-1}$ are identical linear functions supported on the full-dimensional domain $\mathcal{Z}_{\boldsymbol{\theta}}^{L-1}$. It follows that $\mathbf{W}_{L} = \mathbf{W}_{L}'\mathbf{P}_{L-1}^{-1}\mathbf{M}_{L-1}^{-1}$ and $b_{L} = b_{L}'$. $\qquad\square$

