# OpenReview forum: "Functional vs. parametric equivalence of ReLU networks"
_ICLR.cc/2020/Conference — Accept (Poster)_

### Official Review · AnonReviewer1 · 2019-10-21
**Official Blind Review #1**

**Rating:** 3

**Review:**


In this paper, the authors studied the equivalence class of ReLU networks with non-increasing weights, and proved that permutation and scaling are the only function preserving weight transformations. The proof technique is novel, and provides some insights in the geometry space of the loss surface. I think the proof technique could have its general implications to some other research, however, the direct value of this paper is not very clear.

1)	The paper starts with the discussions on the redundancy introduced by over-parametrization, as one of its motivations. However, what is discussed in this paper is actually far distant from over-parametrization. The redundancy in over-parametrized networks and the redundancy in ReLU networks are different concepts and the connection between them is not well established. The over-parametrization is talking about the smooth information flow brought by wide intermediate layers, however, the equivalence class in this paper is more about the mathematical properties of the activation functions and the topological connections of the neural networks. I feel that the authors have some confusing understanding on these two concepts.

2)	The theory was established regarding restrictive types of ReLU networks (feedforward, with non-increasing width). However, many widely used networks are not of this kind. Without extensions to other shapes of the networks, convolutional and recurrent structures, and many kind of normalization transformations (e.g., batch norm and layer norm), the practical value of this paper is limited.

3)	Some important references are missing. The following paper has in-depth theoretical analysis on ReLU networks, and characterizes the redundancy brought by positive-scale invariant properties of ReLU networks. It is strange that the authors did not cite it.  It would be necessary for the authors to discuss their additional technical contributions given this ICLR 2019 paper.
-  Q. Meng, et al. G-SGD: Optimizing ReLU Neural Networks in its Positively Scale-Invariant Space, ICLR 2019

4)	The practical implication of the theories in this paper is not clearly discussed. What if we know there are only these two kinds of redundancies? What kind of new algorithms and practices can be inspired by such theoretical understandings.


** I read the author rebuttal. Different people may have different criteria on evaluating a paper and my criterion is not only about "what", but more importantly about "so what". I still think the authors should think harder about the implications of their work, either theoretically or practically. Furthermore, I understand that some published papers also use restricted settings to ease their proofs, however, I personally do not think this is well justified. I think for top conference like ICLR, more solid and less restrictive works are preferred.  I could at most adjust my score to weak reject.

**Experience Assessment:**

I have published in this field for several years.

**Review Assessment: Checking Correctness Of Derivations And Theory:**

I assessed the sensibility of the derivations and theory.

**Review Assessment: Checking Correctness Of Experiments:**

N/A

**Review Assessment: Thoroughness In Paper Reading:**

I read the paper at least twice and used my best judgement in assessing the paper.

---

> ### Author Response · Authors · 2019-11-10
> **Response to Review #1**
>
> Thank you for your detailed feedback.
> However, we believe that your assessment is based on a serious misunderstanding. We would like to rectify this, and we are glad that the OpenReview system gives us the opportunity to start a dialogue on this.
>
> From your comments, we have the impression that you're an expert in deep learning algorithms and optimization. However, our manuscript is not trying to make a contribution to optimization or learning algorithms, but to the study of mathematical properties and the representational power of deep networks. As we discuss in our detailed reply below, the points you raise -as valid as they are from an optimization perspective- are in no way contradictory to our submission making a valuable contribution to the field of deep network theory.
>
> In light of this clarification, we ask you to reconsider and raise your overly critical score, or at least adjust the self-assessment of your experience. We do not doubt at all your experience in your field, but this is not the field of our submission. Otherwise, we expect your criticism to destroy the manuscript's chances of acceptance, despite the otherwise positive reviews.
>
> **********************************************
> Detailed comments:
>
>
> 1) "over-parametrization"
>
> Clearly, there are many ways in which a network can be "over-parametrized". When studying the optimization properties of deep networks, over-parametrization indeed typically relates to "wider-than-necessary" intermediate layers, as this seems to help convergence to a good solution. That's why we also mention it in the manuscript. But fundamentally, over-parametrization simply means that many parameter choices yield identical functions, and in the manuscript, we use it in this sense. We will upload a revision in which this point is hopefully clearer.
>
>
> 2) "restricted architecture"
>
> Indeed, our results hold for feed-forward ReLU networks with non-increasing width but of arbitrary depth. This is, in fact, a quite broad class of networks for which to establish a mathematically rigorous theoretical result as we do. Proving results about deep network architectures is notoriously hard, and many other theoretical studies restrict themselves to much smaller classes of network architectures, e.g. networks with only a single hidden layer, "linear deep networks" that have no non-linearity between layers, or they establish results not for any fixed architecture but characterize the limit behavior, e.g. when the layer width tends to infinity.
>
> Of course, in practice, many other architectures are used, and it would indeed be great to prove similar results as ours for these. That would go far beyond the scope of an ICLR submission, though. We hope, however, that the proof techniques we introduce in our submission will carry over to some of these architectures (at least convolutional and recurrent ones), thereby laying the foundations for follow-up work.
>
>
> 3) missing reference to [Meng et al, ICLR 2019]
>
> Thank you for pointing us to this reference. We'll be happy to include and discuss it. Its content is completely orthogonal to our work, though: Meng and co-authors build on the indeed well-known fact that ReLU networks are positively scale-invariant and they construct a vector space that is rich enough to represent ReLU networks, yet is positively scale-invariant itself. This allows for easier/better optimization. This is indeed a very nice and elegant result. However, it does not answer or even discuss if there exist *other* symmetries than positive scale-invariance and permutation-invariance, which is the question we study and solve.
>
> As a remark: In Section 11.2, Meng et al use the term "over-parametrized" in essentially the same way we do: the same function can be obtained by different choices of parameters (here: signs of skeleton weights).
>
>
> 4) practical implications
>
> We do not see immediate practical applications of our result, as it essentially is a negative one: no other symmetries exist. But we do not agree that this would in any way diminish our contribution.
>
> Mathematical results should be judged on their own merit, by the insights they provide and the future work they inspire, not by their immediate practical usefulness. Our results prove a mathematical fact about the representation power of deep ReLU networks that had been a long-standing open question before. Our analysis also provides new insights into the space of functions that deep networks can or cannot represent, and our proof techniques provide new tools to the theory community for studying other properties of ReLU networks.

---

> ### Author Response · Authors · 2019-11-13
> **Clarification of "over-parametrization"**
>
> Following up on your feedback, we have uploaded a revision of the paper in which we have hopefully clarified the intended meaning of "over-parameterization".

---

### Official Review · AnonReviewer3 · 2019-10-22
**Official Blind Review #3**

**Rating:** 8

**Review:**

This paper proves that, modulo permutation and scaling, ReLU networks with non-increasing widths are uniquely characterized by the function they induce (excepting some degenerate cases).  This result is not apriori obvious and is of interest.

Authors are commended for balancing brevity, intelligibility, and precision.  However, it is not clear which elements of the proof technique are inapplicable to leaky ReLUs.  It would be helpful to include a brief discussion on this.

I recommend acceptance, since I didn't find any proof errors and the contribution is clear.


**Experience Assessment:**

I do not know much about this area.

**Review Assessment: Checking Correctness Of Derivations And Theory:**

I assessed the sensibility of the derivations and theory.

**Review Assessment: Checking Correctness Of Experiments:**

N/A

**Review Assessment: Thoroughness In Paper Reading:**

I read the paper thoroughly.

---

> ### Author Response · Authors · 2019-11-10
> **Response to Review #3**
>
> Thank you for your feedback!
>
> Following your suggestion, we have updated the manuscript to include a discussion of the applicability of the proof to leaky ReLU (page 8; end of Section 6).

---

### Official Review · AnonReviewer2 · 2019-10-23
**Official Blind Review #2**

**Rating:** 6

**Review:**

The paper shows that for ReLU networks satisfying certain conditions, the weights and biases leading to the exact functional form are the ones obtained by neuron permutations and rescalings, and that no other reparametrizations preserving the function exist. The authors provide an explicit algorithm applying these symmetries.

Disclaimer: My knowledge in this particular subfield is limit and I therefore cannot assess the novelty of the approach.

I find the topic very interesting and the authors’ approach reasonable. I appreciate that they clearly qualify the assumptions used. The apparent tensions between the intuition that many different reparametrizations can exist and their result suggesting that in fact only very few weight space points lead to the same function is also discussed in the conclusion, which I really appreciate.

It would be interesting to study the regime where the functions are not exactly the same. In particular, this is very relevant because we only care about answers on a discrete set of points (train set / test set), and on top of that we only care about the argmax of the logits, rather than the actual detailed answer. I believe such a result would be significantly stronger and more relevant to the practical applications of DNNs, however, I understand that it might be more difficult to obtain.

Overall, I enjoyed this paper and I think it deals with an important problem.


**Experience Assessment:**

I do not know much about this area.

**Review Assessment: Checking Correctness Of Derivations And Theory:**

I assessed the sensibility of the derivations and theory.

**Review Assessment: Checking Correctness Of Experiments:**

N/A

**Review Assessment: Thoroughness In Paper Reading:**

I made a quick assessment of this paper.

---

> ### Author Response · Authors · 2019-11-10
> **Response to Review #2**
>
> Thank you for your feedback!
> We agree that the setting where the functions are not exactly the same is very interesting and important.
> For the reasons you mention, we keep this problem for future work for now.

---

### Public Comment · ~Thiago_Serra1 · 2019-10-11
**Fold-sets, hyperplane arrangements, and union of polyhedra**

I like the direction followed by the authors, but I would like to point out that what you are defining as fold-sets is already known by other names in the literature. In the simpler case of a shallow network, this is a hyperplane arrangement. More generally, this is the union of polyhedra covering the input space. See, for example, Theorem 2 of Raghu et al. (2017), which is further expanded in Theorem 20 of https://arxiv.org/abs/1711.02114

In Figure 2 of the arxiv paper mentioned above, we also identify how the activation hyperplanes of subsequent layers look very different in the input space. In Figure 3 (d), we show that these boundaries for a particular neuron can in fact be disconnected.

---

> ### Author Response · Authors · 2019-10-13
> **Reply to comment**
>
> Thank you for your comment. Yes, fold-sets appear in previous work and we do not claim this definition to be novel or original. Fold-sets do play a central role in our approach, however, and it is useful to have a short name for them. But you are completely right that the fold-set is nothing else as "the boundaries between linear regions" or "the complement of the union of polyhedra" (and that it is a hyperplane arrangement for one-layer nets).
>
> We also thank you for linking the work (Serra et al, 2018); it is clearly relevant and we will reference it.

---

### Decision · Program_Chairs · 2019-12-19

**Decision:**

Accept (Poster)

**Comment:**

This work proves that the weights of feed-forward ReLU networks are determined, up to a specified set of symmetries, by the functions they define. Reviewers found the paper easy to read and the proof technically sound. There was some debate over the motivation for the paper, Reviewer 1 argues that there is no practical significance for the result, a point that the authors do not deny. I appreciate the concerns raised by Reviewer 1, theorists in machine learning should think carefully about the motivation for their work. However, while there is no clear practical significance of this work, I believe there is value to accepting it. Because the considered question concerns a sufficiently fundamental property of neural networks, and the proof is both easy to read and provides insights into a well studied class of models, I believe many researchers will find value in reading this paper.